# Imagined otherness fuels blatant dehumanization of outgroups
Austin van Loon[1] ✉, Amir Goldberg[2] & Sameer B. Srivastava[3]

Dehumanization of others has been attributed to institutional processes that spread dehumanizing norms and narratives, as well as to individuals' denial of mind to others. We propose that blatant dehumanization also arises when people actively contemplate others' minds. We introduce the construct of imagined otherness—perceiving that a prototypical member of a social group construes an important facet of the social world in ways that diverge from the way most humans understand it— and argue that such attributions catalyze blatant dehumanization beyond the effects of general perceived difference and group identification. Measuring perceived schematic difference relative to the concept of America, we examine how this measure relates to the tendency of U.S. Republicans and Democrats to blatantly dehumanize members of the other political party. We report the results of two pre-registered studies—one correlational ($N = 771$) and one experimental ($N = 398$)—that together lend support for our theory. We discuss implications of these findings for research on social boundaries, political polarization, and the measurement of meaning.

Upholding human life as sacrosanct is key to maintaining order in modern Western societies. Indeed, such atrocities as genocide[1–3], racially motivated lynchings[4–6], and ethnic cleansing[7–9] are often rooted in people's tendency to dehumanize others—that is, to perceive members of different social groups in ways that deny their full humanness[10,11]. Even when it does not lead to violent action, dehumanization can still fuel such negative outcomes as the stigmatization of people with mental illness[12], expressions of hostile emotions toward outgroup members[13], and the support of exclusionary immigration policies[14]. Moreover, dehumanization does not only occur in the context of long-simmering racial and ethnic divisions: A growing body of work examines how dehumanization can arise even in the comparatively lower stakes context of political partisanship[15–19]. Across these disparate domains, how do people come to withdraw the dignity of humanity from others?

Sociological research has focused on blatant forms of dehumanization, such as the use of epithets that equate human groups to non-human animals, such as vermin or apes or to less-than-human categories, such as savages. This work has accounted for variation in people's tendency to engage in violent acts or mass atrocities by examining the role of institutions and media in diffusing dehumanizing norms and narratives, as well as the impact of social network ties between groups and the availability of economic resources in blunting these effects [e.g.,[20–22]]. The more subtle, cognitive underpinnings of blatant dehumanization have, however, been largely overlooked in this line of work.

A parallel but mostly separate literature in social psychology has considered dehumanization in both its blatant and subtle forms—for example, failing to attribute distinctively human emotions such as love, hope, and contempt to members of another group [e.g., ref. 23–25]. This work has explained variation in people's tendency to dehumanize others by considering such diverse factors as personality traits, political ideologies, group threat, and intergroup status differentials[26]. Yet a common theme in this literature is that dehumanization—whether blatant or subtle—fundamentally involves the denial of mind to others[27,28]. In this view, dehumanizing others entails either deliberately suppressing full consideration of their minds or passively failing to do so[29,30].

Building on these accounts, we propose that blatant dehumanization also arises through the subtle process of actively considering others' minds. In particular, we argue that merely thinking that members of another group construe the social world in atypical ways can (subject to scope conditions we detail below) catalyze blatant dehumanization. Building on the growing body of work in cultural sociology on schemas[31–35], we introduce the construct of *imagined otherness*—perceiving that a prototypical member of a social group construes an important facet of the social world in ways that diverge from the way most humans understand it. Imagined otherness, we contend, arouses blatant dehumanization above and beyond the effects of general perceptions of difference and group identification. In other words, blatant dehumanization can emerge not only from failing to consider others' minds but also

[1]Duke University, Durham, NC, USA. [2]Stanford University, Stanford, CA, USA. [3]University of California, Berkeley, Berkeley, CA, USA.
✉ e-mail: austin.vanloon@duke.edu

from contemplating how others think about the world and believing their thinking to be aberrant.

To test this idea, we build on the tradition of using explicit association tasks to measure the schemas that exist in people's minds [e.g.,[33]]. We adapt this methodology to focus on a single concept and extend it to measure the schemas that individuals attribute to different social groups and to the typical human. Applying this technique to a central basis of group identity in the United States, the very notion of America, we first demonstrate that the various distance measures it yields are correlated with group identification and intergroup sentiment in the manner one would expect. Next, we report the results of two pre-registered studies that assess the semantic associations about America that self-identified Republicans and Democrats attribute to members of the other political party and to the typical human. Our first study demonstrates that imagined otherness is positively associated with a widely used measure of blatant dehumanization[36] above and beyond the effects of general perceptions of difference and group identification.

Although we theorize that imagined otherness begets blatant dehumanization, our first study is correlational and does not allow us to rule out potential reverse causality—that blatantly dehumanizing an outgroup prompts people to perceive differences in the schemas they attribute to the outgroup versus the typical human. We therefore report the results of a second, experimental study, in which we manipulated perceived semantic difference between members of the opposing political party and the typical respondent on the same categorization task related to America. Results from this experiment establish a causal link between imagined otherness and blatant dehumanization. Together, the two studies suggest an additional cognitive mechanism that fuels blatant dehumanization and thus also point to avenues for curbing it that have not previously been considered. We discuss the implications of these findings for research on social boundaries, political polarization in the U.S., and the measurement of meaning.

A starting point for the development of our theory is the recognition that research on dehumanization has emphasized its role in the choices people make to participate in such atrocities as genocide and sanctioned massacres. Sociologists have tended to focus on its blatant manifestations, examining how institutional processes diffuse dehumanizing norms and narratives through a population[20,21,37]. Hagan and Rymond-Richmond[21], for example, describe how the Sudanese government's efforts to aggregate and amplify racial epithets fueled collective dehumanization and contributed to genocidal victimization in Darfur. In a similar vein, mass media networks are central to Myers'[22] explanation for the diffusion of racial rioting in the United States between 1964 and 1971.

In the context of the Rwandan genocide in 1994, Luft[38] identifies three mechanisms that account for variation in the tendency of Hutus to have engaged in violence directed at Tutsis: transactional (i.e., having sufficient economic resources to resist the pressure to participate); relational (i.e., having social network ties with Tutsis); and cognitive (i.e., engaging in acts of violence that have a reciprocal effect on the tendency to dehumanize). Although this work represents a conceptual integration of institutional and cognitive processes, dehumanization as a phenomenon and its subtle cognitive origins remain relatively understudied in Sociology.

In parallel, and largely disconnected from sociological research, social psychologists have developed four main theories to explain dehumanization[26]: (a) infrahumanization, (b) the dual-factor model, (c) stereotype content, and (d) dementalization. These theories vary in several ways, but pertinent to our theoretical intervention is whether the category of human—to which groups may be denied inclusion—is defined on the basis of traits or the presence of a mind. The traits-based approach to defining humanity, exemplified by the infrahumanization[23,24] and dual-factor[39–41] approaches, posits that groups are denied humanity when they are thought to lack some culturally defined set of traits.

The other approach is exemplified by research on stereotype content[27,28,42] and dementalization[29,43,44]. This perspective argues that the special moral status granted to humans is not simply the result of whether a group is thought to possess a checklist of characteristics but is fundamentally intertwined with the human faculties of social perception. Specifically, when

we encounter another person in a social setting, we deploy a theory of mind through which we reason about their internal states[45]. To dehumanize a group, according to this viewpoint, is to not engage this fundamental form of social cognition on behalf of group members—that is, to deny them the existence of a mind, the perspective of which can be taken by another.

Building on these ideas, we argue that dehumanization arises not only from believing that others have an impoverished mind, but also when one thinks that others have an inherently different mind. In particular, we propose that it occurs through people's lay—and often unselfconscious—theories about inherent and specific differences between the worldviews of others and those of typical humans. Once members of another group are believed to be interpreting the world in ways that are antithetical to how most people do so, their perceived humanity erodes. We refer to this process as *imagined otherness*.

Support for this argument comes from research by cognitive and social psychologists, which demonstrates that people are innately disposed to attribute differences between social categories to deep, immutable, and mostly unobservable distinct essences[46,47]. These essentialized differences arise in subtle ways and often persist even when they are normatively rejected. For example, despite the erosion in recent decades of interracial boundaries in some parts of the U.S., white Americans continue to see people of other races as inherently different from themselves[48].

We argue that such essentialized processes of perceived group difference also fuel dehumanization. Building on Khazzoom's[49] insight that exclusion requires the articulation of group difference, we propose that a specific form of articulation—imagining how members of a different group see the social world relative to how a typical person regards it—can lead to explicitly denying them full humanity. We anticipate that the greater the divergence in these two imagined perceptions, the more likely a person is to dehumanize members of another group—even when accounting for general perceptions of difference or the strength of group identification.

Under what conditions should we expect this process to unfold? One important scope condition, we propose, is whether the prototype of most people serves as a reference point for legitimacy. While normative expectations shape the behavior of most social groups[50], some groups may focus less on the expectations of others and more on systems of morality that are core to their identity. Potential examples might include hate groups such as the Ku Klux Klan, religious communities such as Amish Mennonites, and cults. For such groups, we conjecture that perceiving an outgroup as being different from most people may not give rise to dehumanization because the category of most people is not necessarily valued. Similarly, if an outgroup is viewed as elite or especially enlightened, we might expect that perceptions of its difference relative to most people need not produce dehumanization given that the outgroup may be viewed as more legitimate than the category of most people.

What does it mean to imagine that a person sees the social world in fundamentally different ways from another person or a typical human? Social scientists commonly make inferences about how people understand the world by asking them about their attitudes. Attitudes, however, do not fully capture the meaning structures undergirding people's understandings of themselves and others[31,33,51]. Working-class Londoners, for example, appreciate material excess as much as those higher on the income distribution do. However, when they describe their mentality as inherently different from that of posh stuck-ups, they allude to a self-perceived authentic ordinariness, contrasted with the contrived propriety of the upper classes, that cannot be captured by a single attitude [ref. [52], p. 152]. In fact, the same normative attitude can often be the product of different interpretations. Most Americans, for instance, value hard work. But when rural Wisconsinites describe their belief in hard work as something that distinguishes them from city dwellers, they invoke an appreciation of toughness that, as they see it, is neither understood nor valued by their fellow citizens in large cities[53].

These archetypal descriptions of social groups relate to different ways by which people construct meaning. An extensive body of work demonstrates that people represent meaning as semantic relationships between

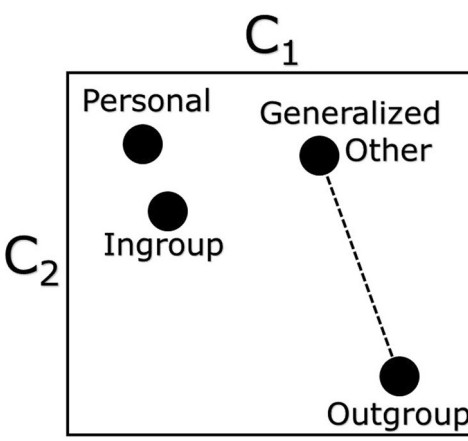

**Fig. 1 | Graphical representation of schematic differences.** Graphical representation of schematic representations in the mind of a hypothetical individual. Dots represent personal, ingroup, outgroup, and generalized other schemas of a focal concept. A schema's location corresponds to its level of association with two other concepts, $C_1$ and $C_2$. The dashed line represents the schematic distance of interest, which we refer to as *Generalized Outgroup Distance*.

concepts[51,54–56]. By concept, we mean an abstract mental representation of a feature of the world. When rural Wisconsinites talk about hard work, they invoke such a concept. The meaning they assign to hard work is implicit in the cognitive associations this concept has with other concepts, such as toughness. Accounting for these cognitive associations helps uncover the meaning structures that underlie people's beliefs. Americans who benefit from the market, for example, tend to express unbridled faith in capitalism. Nevertheless, the various ways by which the economically advantaged perceive associations between different market activities suggests that this faith is fueled by different understandings of the moral boundaries of markets[57].

Imagine two Americans who are self-professed believers in freedom. Whereas a conservative might associate freedom with economic independence and the lack of government intervention, a liberal might associate it with civil liberties and the freedom of expression[58]. Drawing on conventional sociological nomenclature, we refer to these different associations, and the meanings they embody, as *schemas*. Note that schemas are generally conceptualized as the network of associations between concepts that people hold in their minds[31,33]. That said, the term has been used in such a variety of ways that it has almost become an omnibus term for all cognitive representations[35]. Although our conceptualization maps more closely to the construct of construals[59], or mental representations about a given domain, we retain the more widely used term, schemas, to follow the convention in the literature. As a growing body of research by cultural sociologists demonstrates, members of different groups often make associations about the same concept in systematically different ways. People with differing class, political, or religious identities have divergent schemas about issues as varied as what it means to be American[60], the role and epistemological authority of science[61,62], and the structure and social prestige of occupations[63].

This prior work explores the different ways that members of social groups interpret reality. We refer to these as differences as their *personal schemas*. In contrast, our theory pertains to the schemas that an individual attributes to others. We theorize about two such schemas. The first involves imagining how members of different groups perceive the world, which we refer to as the *outgroup schema*. The second builds on the insight that individuals also have a mental representation of persons in general. As prior work demonstrates, people imagine what most others expect as they determine whether and how to adjust their behavior to conform to these generic expectations[64,65]. Importantly, thinking about this generalized other is not the same as simply constructing a composite image based on a weighted average of known groups. Rather, it represents a distinct cognitive

act—one that people are capable of doing when, for example, thinking about specific political parties and then abstracting to typical members of society[66]. Drawing on Mead's[67] nomenclature, we refer to this mental representation as the *generalized other schema*.

How do the outgroup schema and the generalized other schema that people imagine in their minds jointly relate to dehumanization? Consider the stylized individual mental representation illustrated in Fig. 1. This model depicts a hypothetical individual's *personal, ingroup, outgroup,* and *generalized other* schemas of a concept $C$ (e.g., freedom). The dimensions of this conceptual space correspond to two other concepts, $C_1$ (e.g., governmental regulation) and $C_2$ (e.g., civil liberties). For ease of presentation, we illustrate only two dimensions—although in reality a cognitive conceptual space is hyperdimensional. A schema's location in this space relates to the level of association between the focal concept $C$ and other concepts. Thus, schemas that are closer together in this space have more shared associations between them.

By positioning schemas in such a space, we can measure schematic distances *between people*. For example, two people who hail from individualistic cultures may have personal schemas about freedom that are closer together than are the personal schemas of two individuals who come from different national cultures—one individualistic and one collectivist. Yet because people have not only personal schemas but also schemas they attribute to others, we can also derive *within-person* measures of perceived schematic differences. For example, we can assess the distance between a person's personal schema and her ingroup schema. We might expect, for example, that a person who identifies strongly with her ingroup will have a personal schema that is closer to her ingroup schema than to her outgroup schema.

Applying this conceptual apparatus to our outcome of interest, we anticipate that outgroup dehumanization is related to the relationship between a person's outgroup schema and her generalized other schema. In particular, we posit that the tendency to dehumanize an outgroup will increase with the distance between a person's outgroup schema and her generalized other schema, which we refer to as *generalized outgroup distance* and which is represented in Fig. 1 by the dashed line. Although our motivating example focuses on the concept of freedom, we recognize that our theory is unlikely to extend to associations about concepts with little symbolic meaning—for example, chair or gerbil. We conjecture that concepts that are either relevant to ingroup and outgroup identities or that have otherwise acquired symbolic value are ones that can activate our proposed imagined otherness mechanism.

Perceptions of difference can, of course, also lead to negative affect about an outgroup. Yet prior work has shown that dehumanization and disliking are analytically distinct phenomena[25]. For instance, one may very much enjoy the company of pets but also understand that they are not human. More work needs to be done to fully understand the relationship between antipathy and dehumanization, and we therefore remain agnostic about the relationship between generalized outgroup distance and the affective evaluation of a group. Based on with social identity theory[68], we anticipate that perceiving an outgroup schema as distant from one's personal schema will lead to dislike of the outgroup—an idea we test (with the results reported in Supplementary Note 2) to help validate our measurement strategy. That said, the arguments above lead to the following pre-registered hypothesis:

**Main hypothesis**

*The tendency to exhibit outgroup dehumanization will increase with perceived generalized outgroup schematic distance.*

The pre-registrations for our correlational and experimental studies can be found at https://osf.io/kba36 and osf.io/5t79a, respectively. The correlational study was pre-registered on January 8, 2021. The experimental study was pre-registered on May 17, 2021. In this paper, we focus on what is referred to as Hypothesis 5 in the first pre-registration and Hypothesis 2 in the second pre-registration. Imagined otherness corresponds to the variable referred to as generalized-outgroup construal distance in the

pre-registrations. MCET in the pre-registrations refers to our schema elicitation task.

To test our theory, we apply it to the context of political identity in the United States. By many measures, the American political sphere has become increasingly polarized in recent decades[69,70], especially among political elites[71,72]. While there is disagreement as to how much ordinary Americans have grown apart in their ideological and policy preferences[73,74], there is little doubt that Republicans and Democrats have become increasingly antagonistic toward members of the other party[75,76]. The independent effects of growing interparty hostility are difficult to isolate; however, multiple indicators suggest that it has been increasing as American political discourse has become more divisive [e.g.,[77]] and Americans' faith in democratic institutions has declined [e.g., ref. [78]].

Partisan politics is a useful context for exploring the dehumanizing implications of imagined otherness for two main reasons. First, unlike race or ethnicity, party identity is not ascribed at birth. Moreover, partisan affiliations can often change significantly throughout one's lifetime. It is therefore likely that, to the extent that it exists, party-based imagined otherness is not anchored in lay theories of biological difference. Second, recent work has already established that some but not all Americans see members of the opposing political party as less than human[16]. We contend that variation in this tendency is partly rooted in the belief that members of the other party see the world in inherently different ways.

Sociologists have recently begun to examine differences in schematic understandings across party lines. Hunzaker and Valentino[33], for example, find that Democrats and Republicans have different interpretations of poverty, while Baldassarri and Goldberg[79] show that schematic heterogeneity crosscuts political identities. These studies examine differences in how members of different parties understand the world. Our primary focus, in contrast, is perceived schematic distance: individuals' beliefs about opposing party members' understandings of the world. Building on our theoretical framework, we argue that perceived generalized outgroup schematic distance is related to opposing party members' dehumanization.

To test our theory, we apply it to Democrats' and Republicans' group construals of America. We expect that, when people imagine that typical outgroup members' schemas of America differ from those of the typical person, they are less likely to see them as fully human. Previous research has convincingly shown that the American identity is contested[60,80,81]—and especially so across party lines[82,83]—yet we know little about how partisans imagine those on the other side of the aisle understands this shared identity. America is an ideal stimulus for us to test our theory with, since it meets both of the scope conditions we mentioned above. First, America is a relevant concept to the identity of Democrats and Republicans. Second, we expect most participants will believe that the prototypical Republican, Democrat, and person would value this concept. Our choice of America as a stimulus, however, may provide a conservative test of our theory, since Levendusky[84] demonstrates that, when primed to think about themselves as Americans, interparty hostility between politically affiliated Americans declines.

One question that naturally arises is whether outgroup dehumanization is simply an expression of outgroup antipathy. Indeed, in our empirical setting, we find that affective polarization—a commonly used measure of the degree to which partisans prefer their own party over the other party[85]—is strongly correlated with outgroup dehumanization (in our correlational study, $r = 0.54$). However, previous work has demonstrated that blatant dehumanization is not tantamount to disliking an outgroup—whether in intergroup contexts generally[36] or the specific domain of politics[16]. Additionally, as mentioned above, our theory is agnostic both to the relationship between imagined otherness and outgroup prejudice as well as the relationship between outgroup prejudice and dehumanization. For these reasons, we do not consider affective polarization further in the results discussed below.

Before describing our methods, we first review prevailing approaches for measuring schemas and explain why they are ill-suited for our purposes. Existing approaches to measuring schemas fall broadly into two categories: relational transformation techniques and association tasks. Relational transformation techniques rely on traditional survey responses: Respondents are asked to provide their attitudes on independent concepts. Measures of the strength of association between these attitudes (e.g., correlation[86] or relationality[31]) are transformed into inter-attitude and inter-respondent distances. Relational transformation techniques are useful in identifying subsets of respondents who organize meaning in distinctive ways and can be applied to data that were not specifically designed to elicit cultural schemas. Because they are designed to measure attitudes, however, schematic representations can only be inferred indirectly.

In contrast, association tasks directly probe the cognitive associations between concepts, either by asking about them explicitly [e.g.,[33]] or eliciting them implicitly [e.g.,[51]]. Explicit association tasks typically suffer from two problems. First, they tend to conflate interpretation with valence. In the Concept Association Task used by Hunzaker and Valentino[33], for example, respondents are asked to associate multivocal concepts such as immigrant with value-laden concepts such as lazy or dishonest that usually have strong positive or negative connotations. Respondents who report such associations and who do not may differ in at least two ways. One is that they may genuinely interpret the category of immigrant differently—for instance, one subgroup might think of low-skill immigrants, while another might imagine high-skilled immigrants. Another possibility is that they feel different levels of warmth towards the category of immigrants and thus report different levels of association between this category and lazy, dishonest, or any other clearly positive or negative concept put before them. The design of the Concept Association Task makes it impossible to differentiate between these possibilities. The second problem with explicit association tasks is that, because associations of value-laden concepts are included, responses are susceptible to social desirability bias. For example, respondents might feel that associating immigrant with lazy is normatively inappropriate, even if they would espouse such an association in private.

To overcome social desirability bias, implicit association tasks typically use response latencies to assess the degree to which respondents differentially associate two pairs of concepts such as black and white with pleasant and unpleasant[87]. Accurate assessments require, however, that respondents complete dozens of trials for a given concept pair. Given that we seek to assess a broad range of associations that people attribute to outgroup members and the generalized other, an implicit association task would be unwieldy for our purposes and would likely result in participant fatigue and low-quality responses.

## Methods

To test our hypothesis, we recruited a total of 1,169 individuals from Prolific to complete two pre-registered studies that were approved by the Stanford Institutional Review Board. The first was a correlational study in which participants provided informed consent before participating. The second was a survey experiment, in which participants were temporarily deceived but were informed of this deception (and given the opportunity to withhold their data) after participation. All relevant ethical regulations were followed in conducting this research.

Evaluating our main hypothesis requires us to elicit the schematic representations of a target concept that participants attribute to outgroup members and to the generalized other. We describe below the technique we used, which builds on the prior methods described above but addresses some of their limitations.

### Measuring generalized outgroup schematic distance

To address the limitations of existing methods identified above, we used our own, modified version of a schema elicitation task. Building on the logic of the well-established pile sort method[88,89], our instrument uses multiple forced-choice categorizations to tap into participants' schematic representations of a focal concept. In the course of completing the instrument, participants are asked to categorize subsets of *associated concepts* relative to a *target concept*. Although our particular implementation focused on the target concept of America, this approach can be readily adapted to other target and associated concepts [see, for example,[90]].

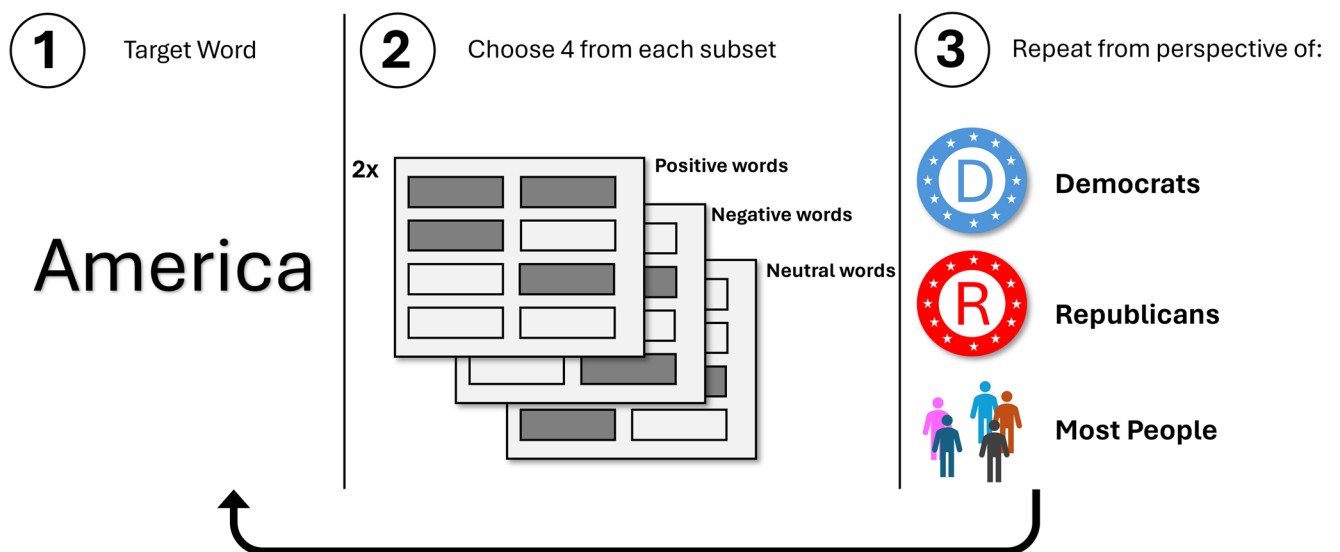

**Fig. 2 | Schema elicitation procedure.** Illustration of the schema elicitation procedure, applied to the concept America and the context of partisan politics.

For a given subset of associated concepts, participants are presented with eight words and are asked to identify four of these eight that most belong in a category with the target concept. By assessing participants' choices across several such subsets, our instrument efficiently taps into the many cognitive associations about a target concept that participants either make themselves or attribute to others. It also uncovers the potential associations that participants do not make or do not attribute to others. Thus, it yields a profile of concepts that are either associated or not associated with the target concept.

We configured the instrument such that participants were shown six subsets of associated concepts: two with only positive words, two with only negative words, and two with only neutral words. To avoid order effects, we randomized the order in which these sets were presented to participants, as well as the position of each word in each set. The division of the task into fixed-valence subsets allowed us to distinguish respondents' interpretations of the target concept from their sentiment toward it. This feature of the instrument also helped neutralize the threat of social desirability bias by forcing all respondents to associate the target concept with the same number of negative, positive, and neutral associated concepts.

A question that arises about this measurement strategy is whether people are able to make associations that reflect their schematic understanding of a focal concept when they harbor strongly positive or negative sentiments about that concept. For example, if a respondent feels unambiguously positively about America, to what extent can she reliably associate negative terms with this concept? Given that both Republicans and Democrats tend to express pride in different facets of the country while also harboring grievances about the current state of affairs, we think this issue is less concerning in our empirical context.

We configured four versions of the instrument that varied based on the perspective we asked participants to take: (a) the personal perspective, or the categorizations they themselves made; (b) the Republican perspective, or the categorizations they ascribed to a prototypical Republican; (c) the Democratic perspective, or the categorizations they ascribed to a prototypical Democrat; and (d) the generalized other perspective, or the categorizations they ascribed to most people. Our main hypothesis focuses on tasks (b), (c), and (d). We used task (a) for validation checks of our measurement strategy. Depending on the respondent's partisan identity, versions (b) or (c) corresponded to her ingroup and outgroup perspective. In (a), respondents were asked to select the four words "you think most belong in a category with" the target word. This was in (b) altered to "you think Republicans think", in (c) to "you think Democrats think", and in (d) to "you think most people think." Fig. 2 provides an illustration of our procedure.

## Stimuli selection

Our choice of associated concepts was guided by three objectives. First, to separate associations from positive or negative evaluations, we sought to minimize variation in the valence of words within each subset. Because only associated concepts within a subset were compared, we reduced the risk that participants' choices were driven by the associated concepts' valences. Second, we wanted to choose words that could plausibly be associated with America in participants' minds. Finally, in identifying potentially associated concepts, we sought to generate a set of potential words that would resonate with a broad cross-section of U.S. Republicans and Democrats and not encapsulate our own potentially biased perspectives.

To accomplish these aims, we conducted a preliminary study in which we asked participants on Prolific ($N = 200$), a widely used research-oriented participant pool, to make free word associations with America. Specifically, we asked participants to think about America and to provide five generally positive words, five generally negative words, and five neither positive nor negative words that came to their mind. After lower-casing and lemmatizing all generated words, we were left with 955 unique terms. The distribution of frequency over these terms was highly skewed: 582 words were only mentioned once, 51 words were mentioned at least ten times, and one word (freedom) was mentioned over one hundred times. To identify words that could plausibly be associated with America in participants' minds in our eventual main studies, we selected words that were mentioned at least three times by participants in this preliminary study. This left us with 57 positive words, 59 negative words, and 56 neutral words.

These words varied in sentiment even within a given category, and some were also ambiguous in sentiment. Thirty words appeared at least three times in at least two sentiment categories (positive, negative, or neutral). Two words—money and capitalism—appeared at least three times in all three sentiment categories. To quantify each word's typical sentiment and assess its ambiguity, we relied on word embedding models. Specifically, we used three separate embedding models (GloVe GigaWord, GloVe Twitter, and Word2Vec Google News) to assess the sentiment of each word by measuring its cosine distance to a centroid of negative words (bad, negative and immoral) minus its cosine distance to a centroid of positive words (good, positive, and moral). We measured both the mean and the variance of this measure across the three embedding models. We also measured the average cosine distance between each word and the target concept America.

Based on this information about each word's sentiment and its typical association with America, we selected six sets of eight words that: (a) exhibited low within-set variation in sentiment and low cross-corpus variance in sentiment; (b) were within the interquartile range of semantic

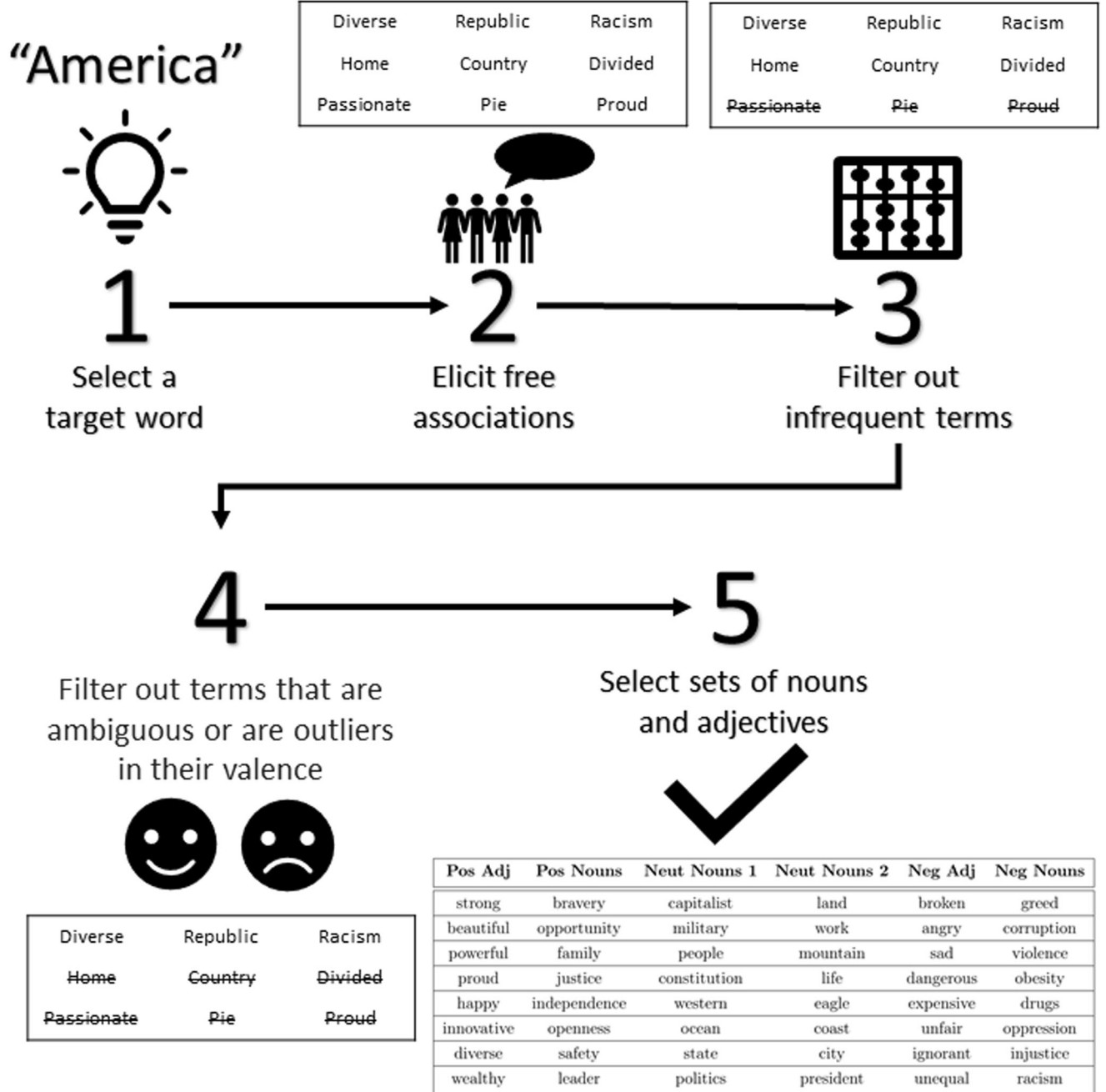

**Fig. 3 | Stimuli selection procedure.** Illustration of the process by which we selected the associated words we used in our version of the schema elicitation task.

association with America; and (c) were invoked by participants at least three times. We were also interested in whether nouns or adjectives would exhibit systematically different patterns, so we maintained sets that were either all adjectives or all nouns according to the word's first listed sense in WordNet[91]. This procedure and the final subsets of words we selected are shown in Fig. 3. Supplementary Table 1 (located in Supplementary Note 1) lists the specific words we selected based on this procedure. In Supplementary Note 2 (which contains Supplementary Fig. 1, Supplementary Fig. 2, and Supplementary Table 2), we also report the results of several additional analyses that help establish the validity of our measurement technique.

### Correlational study

We sought and were mostly successful in recruiting a sample from Prolific that was nearly equally split between Republicans and Democrats and was close to nationally representative in terms of gender (51% Female), race/

ethnicity (76% white, 11% black, 7% Asian, and 6% mixed or other), religious affiliation (60% some type of Christian, 27% Atheist or Agnostic, 2% Jewish, 1 % Muslim, 10% some other religious affiliations), and geographic region of the Unites States (25% Northeast, 21% Midwest, 29% South, and 23% West). Participants, after providing informed consent to participate in the study, were asked to complete the schema elicitation task from their personal perspective first (to acclimate them to the task) and then, in a random order, to fill it out from the perspective of Republicans, Democrats, and most people. Finally, they were asked various attitudinal measures, including our control and dependent variables.

To determine respondents' ingroup and outgroup affiliation, we asked how they typically think of themselves: as a Republican, as a Democrat, as a member of a third party, or as something else. After dropping respondents who (1) completed the survey more quickly than half the median response time (as we pre-registered), (2) did not complete all versions of the schema elicitation

task in full, or (3) did not identify as a Republican or Democrat, we were left with 771 participants—382 of whom (49.5%) identified as Republican.

While studies of dehumanization were focusing on its subtle manifestations[36], introduced the Ascent of Man scale, which measures—in the most blatant terms possible—the degree to which an individual views a group as less-than-human. The version of the scale we used asked participants to rate both Democrats and Republicans on a slider scale from 0 to 100, corresponding to different images depicting the Ascent of Man from apes to fully upright humans. Following prior studies, we defined outgroup dehumanization as a participant's ingroup rating minus her outgroup rating. Higher values of this measure correspond to reporting the outgroup as less human than one's ingroup. The ingroup rating is included in the measure to account for baseline differences in how participants use the dehumanization slider scale, though our measure is highly correlated with just the slider rating given to one's political outgroup; $r = -0.83$.

The use of this scale in the context of U.S. politics raises some important analytical questions (we thank an especially insightful reviewer for raising these concerns). First, the scale appears to assume that participants believe that humans evolved from apes—a belief that may be less popular among political conservatives. Second, the scale depicts a male human, which may conflict with gendered expectations of political preferences[92]. Importantly, both of these issues would suggest differences in the mean level of dehumanization across political parties.

Despite these concerns, we chose to use this scale for three main reasons. First, it has been extensively validated by Kteily and colleagues[36]: It has been shown to correlate with other measures of dehumanization, to be reliable over time, and to be predictive of important attitudes and behaviors towards several groups above and beyond prejudice. Second, it is the most commonly used measure of dehumanization in the political realm to date [e.g.,[16,19]]. Finally, it is efficient to implement given that it asks participants to report their visceral sense of how they view a group.

Our main independent variable is imagined otherness—that is, the degree to which a participant's attributed interpretations of America from the perspective of their political outgroup and from the perspective of most people are divergent. This construct corresponds to generalized outgroup schematic distance illustrated in Fig. 1. Formally, we defined the distance between these two interpretations, labeled $A$ and $B$, using a simple divergence metric:

$$D(A, B) = 1 - \frac{2|A \cap B|}{n} \quad (1)$$

where $n$ is the number of associated concepts (48 in our case) in each version of the schema elicitation task. This measure can be interpreted as one less the proportion of selections in $A$ that are also in $B$, with higher values indicating that $A$ and $B$ are more dissimilar. We mean-centered and standardized this measure (except in reporting descriptive statistics) to ease interpretation.

Our models include a set of control variables that previous literature has related to dehumanization. For example, Martherus and colleagues[16] find that strong party identification is associated with both subtle and blatant dehumanization. Other work shows that conservatives are more likely to espouse authoritarian positions[93], which has been tied to outgroup dehumanization[36]. To account for outgroup dehumanization that is driven by partisanship and ideology, we included the following control variables in our models:

- *Democrat*. Participants were asked whether they identify as Republican or Democrat (recall that respondents with other forms of partisan identifications were removed from the sample).
- *Strong partisan*. Consistent with how the American National Election Studies (ANES) measures strength of party identification, participants were asked after reporting their party affiliation whether they identified as a strong [Democrat/Republican] or a not very strong [Democrat/Republican]. One benefit of controlling for identification strength is that it is likely correlated with the degree of political homophily in participants' networks[94]. Thus, insofar as people's views of the outgroup or the generalized other are informed in part by their exposure to

members of the opposing party, this control variable should at least partially account for such variation.
- *Ideological extremity*. We asked participants to report their political ideology using the ANES 7-point ideology scale and treated each response's distance from the midpoint (moderate or middle of the road) as a measure of ideological extremity. We used this as a continuous measure (although the results we report below are robust to modeling it as a categorical variable).
- *Extreme conservative*. Given the association between authoritarianism and conservatism, we included a dummy variable for extreme conservatism in dehumanization models. This variable was set to 1 for respondents who reported being extremely conservative on the ANES 7-point scale and to 0 otherwise.

While our key independent variable—imagined otherness—corresponds to the divergence between one's outgroup schema and generalized other schema, divergences between other pairs of schemas also plausibly correlate with blatant outgroup dehumanization (see Fig. 1). We therefore compute two additional divergence measures that could be related to blatant dehumanization. The first is the divergence between one's personal schema and one's outgroup schema (i.e., how differently does one respond to "what do *they* think?" and "what do *you* think?"). We refer to this divergence as the *personal-outgroup divergence*. The second is the divergence between one's ingroup schema and one's outgroup schema (i.e., "what do *they* think?" compared to "what do *we* think?"). We call this the *intergroup divergence*. Both are calculated using equation (1), but $A$ and $B$ change accordingly. Both measures are also standardized.

Finally, to assess whether our schema elicitation task taps into schematic differences above and beyond what people would report in response to a simple survey question, we also collected two self-report measures of perceived atypicality of the outgroup's perspective. Specifically, participants were asked on a Likert scale from 1 (Strongly disagree) to 5 (Strongly agree) how strongly they agree that members of their political outgroup: (1) "think about America in a very abnormal way, or a way that is different from how most people think about America"; and (2) "see the world in a very abnormal way, or a way that is different from how most people see the world". We combined these responses into a composite measure ($\alpha = 0.85$).

### Experimental study

We aimed to recruit 200 Republicans and 200 Democrats from Prolific to complete our survey experiment; however, the procedure described below resulted in a sample that came close to but did not precisely achieve this target. As in the correlational study, we asked participants who gave consent to participate in the research about their party affiliation and screened out individuals who did not identify as a Democrat or a Republican. This step occurred prior to random assignment. We then asked participants to report on outgroup dehumanization using the same scale as in the correlational study, as well as to complete the schema elicitation task—but only from their own perspective (i.e., to provide their personal schema about America). We had participants complete the schema elicitation task to put into context the information they would later receive as part of the experiment about distances between outgroup members and typical respondents. Participants were then randomly assigned to one of two conditions. In both conditions, they were shown the following text (with [blue/red] dots and [Democrats/Republicans] selected to correspond to each participant's outgroup):

*"Below is a visual representation of how [Democrats/Republicans] and typical respondents completed the same word association task [that you just completed] for the concept of 'America.' Each dot represents an individual, with [blue/red] dots representing [Democrats/Republicans] and black dots representing typical respondents. The closer two dots are to each other, the more similar were the responses between those two individuals."*

Next came our experimental manipulation, which involved temporarily deceiving participants by showing them a fake visualization of the described results. Per our IRB-approved study protocol, we disclosed the deception at

## Less difference

## Greater difference

Republicans

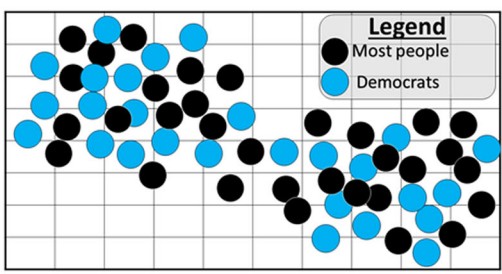 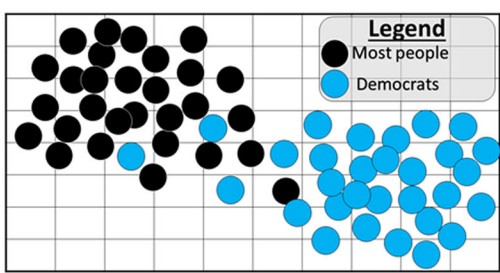

Democrats

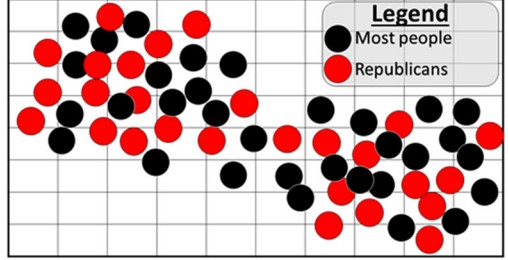 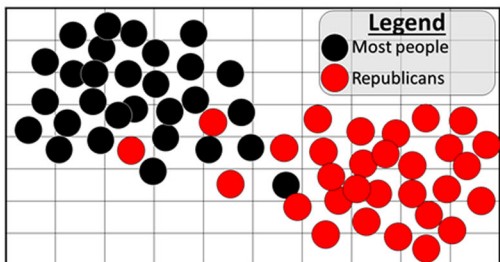

**Fig. 4 | Experimentally manipulating imagined otherness.** Graphic shown to participants depending on their party and their randomly assigned condition.

the conclusion of the study, revealed the purpose of the study, and explained why deception was necessary. In one condition (*N* = 197; 98 Republicans), participants were shown a visualization that suggested members of their political outgroup responded very differently on average from most people. In the other condition (*N* = 201; 102 Democrats), political outgroup members' responses were depicted as mostly interchangeable with the responses of most people. These two images are displayed in Fig. 4. Participants were then asked to report on outgroup dehumanization and were asked additional attitudinal questions. After being debriefed, participants were asked whether we could use their data. One Republican respondent and one Democrat respondent asked us to not use their data and are excluded from analysis. Our final analytical sample contains 197 Republicans and 201 Democrats.

We used one of the self-report measures (i.e., the question about perceived differences between the outgroup and most people in their views of America) as a manipulation check. Specifically, we assessed the change in this self-report before and after participants received the experimental manipulation. We anticipated that a successful manipulation would result in participants who were shown the visualization depicting greater difference between the outgroup and the generalized other reporting a larger increase in their self-reports than participants who were shown the visualization depicting less difference.

Our dependent variable was based on the Ascent of Man scale that we also used in the correlational study. One difference is that we used a repeated measure design (i.e., we measured the *change* in responses to the scale before and after treatment) to increase the precision with which our treatment effect is estimated[95]. Our dependent variable is each participant's post-manipulation value of outgroup dehumanization minus their pre-manipulation value. A value of zero indicates that participants reported the same levels of outgroup dehumanization across the two measurements, while a positive (negative) value indicates that respondents reported more (less) outgroup dehumanization after the experimental manipulation than before. We expected, in accordance with our hypothesis and the results from our correlational study, that participants assigned to the greater schematic difference condition would have higher values on the change measure than participants assigned to the less difference condition.

### Additional information

The data for the correlational study was collected between February 2, 2021, and February 10, 2021. The data for the experimental study were collected between April 5, 2022, and April 26, 2022. Samples for both studies were collected from Prolific for reasons of convenience and cost. The use of this convenience sample may have biased our results in several ways. We sought to recruit only Democrats and Republicans. Both samples are representative based on a set of demographic quotas, but they are likely unobserved characteristics on which this sample is not representative of the population.

Data for both studies was collected using Qualtrics. Participants completed the studies on whichever devices they were using when they signed up for the study on Prolific. Responses were recorded via Qualtrics. In the correlational study, there was no randomization into conditions. In the experimental study, participants were randomly assigned into one of the two conditions by way of a Qualtrics Randomizer in the survey flow.

For the correlational study, participants were dropped if they did not complete the survey (*N* = 40), if they took less than half the median duration to complete the study (*N* = 84), or if they did not identify as a Democrat or Republican (*N* = 2). All criteria were preregistered. In the experimental study, participants were dropped if they did not identify as Democrat or Republican (*N* = 29) or if they did not give consent for us to use their data after being debriefed (*N* = 3). In the correlational study, 31 participants (around 3%) did not complete the survey after starting it. In the experimental study, 5 participants (about 1%) did not complete the study after starting it. We have no way of knowing why these participants attritted.

For the correlational study, gender was used in the sampling of participants. Specifically, we asked Prolific to provide a representative sample along various demographic dimensions including gender (51% identified as female). Gender was collected passively from Prolific after data collection. Participant gender was not collected during the experimental study.

We used Python 3.8.5 to clean, analyze, and vizualize the data. To do this, we made use of the following packages: SciPy (version 1.7.3), Pandas (version 1.5.1), Statsmodels (version 0.13.2), MatplotLib (version 3.6.2), and Seaborn (version 0.11.0). Throughout, data distributions are assumed to be normal but this was not formally tested.

### Reporting summary

Further information on research design is available in the Nature Portfolio Reporting Summary linked to this article.

## Table 1 | Correlational study descriptive statistics

| | μ | σ | 1 | 2 | 3 | 4 | 5 | 6 | 7 |
|---|---|---|---|---|---|---|---|---|---|---|
| 1. Imagined otherness | 0.40 | 0.12 | 1 | | | | | | |
| 2. Outgroup dehumanization | 19.07 | 29.10 | 0.19 | 1 | | | | | |
| 3. Self-reported imagined othering | 1.28 | 1.61 | 0.23 | 0.42 | 1 | | | | |
| 4. Democrat | 0.51 | 0.50 | −0.13 | 0.16 | −0.07 | 1 | | | |
| 5. Strong partisan | 0.56 | 0.50 | 0.03 | 0.32 | 0.30 | 0.22 | 1 | | |
| 6. Ideological extremity | 1.86 | 0.86 | 0.05 | 0.25 | 0.27 | 0.14 | 0.52 | 1 | |
| 7. Extreme conservative | 0.09 | 0.29 | 0.13 | 0.14 | 0.18 | −0.31 | 0.25 | 0.43 | 1 |

## Results

### Correlational Study

Table 1 reports descriptive statistics and Pearson correlations between a selection of the variables used in our analyses. Although the mean of outgroup dehumanization is 19.07, its median value is only 7. Indeed, 36% of participants reported the exact same level of humanity for their outgroup as for their ingroup (26% gave a full humanness rating of 100 to both groups). In other words, these participants did not exhibit any outgroup dehumanization. In our sample, Democrats dehumanize Republicans more than Republicans dehumanize Democrats. Lastly, consistent with previous research, strong partisan identity, ideological extremity, and identifying as an extreme conservative are all positively correlated with outgroup dehumanization.

We report results of OLS regression models predicting outgroup dehumanization in Table 2. Traditional standard errors are presented, but all reported significance levels are consistent with heteroscedasticity-consistent standard errors and bootstrapped standard errors calculated with 10k repetitions. Consistent with our main hypothesis, Model 1 shows that participants who imagined the prototypical outgroup member as having interpretations of America that diverged from that of most people exhibited more outgroup dehumanization. In Model 2, we see that this association is robust to the inclusion of our control variables—including various measures of group identification. Models 3-4 show that, when controlling for imagined otherness, neither of the alternative divergence metrics we calculated significantly relate to outgroup dehumanization. Finally, Model 5 establishes that our hypothesized association is robust to the inclusion of the self-report measure as an additional covariate. Together, these results lend support for our theory. Moreover, they establish that differences in schematic interpretations are related to dehumanization independent of group identification or abstract perceived differences.

Figure 5, corresponding to Model 1 in Table 2, depicts the bivariate relationship between imagined otherness and outgroup dehumanization. Respondents one standard deviation below the mean in imagined otherness are expected to dehumanize their political outgroup by 13.7 points relative to their political ingroup, while respondents one standard deviation above the mean in imagined otherness are estimated to rate their ingroup as 24.5 points more human than their outgroup. For context, Kteily and colleagues[36] find that the ethnic group most dehumanized by Americans—Arabs—are assigned, on average, 13.9 fewer points than Americans are on this same scale.

### Empirical extensions

Although they were not pre-registered, we conducted two supplemental analyses that we believe shed additional light on our findings. The details of these analyses are reported in supplementary note 2. In the first, we explore heterogeneity in the relationship between imagined otherness and outgroup dehumanization. Specifically, we find that divergence between a participant's personal schema and her ingroup schema significantly weakens the relationship between imagined otherness and outgroup dehumanization. In other words, the tendency for imagined otherness to be associated with outgroup dehumanization is stronger for those whose personal schema is

## Table 2 | OLS models of outgroup dehumanization

| | (1) | (2) | (3) | (4) | (5) |
|---|---|---|---|---|---|
| Imagined otherness | 5.40*** | 5.38*** | 5.24*** | 5.19*** | 2.65** |
| | (1.03) | (0.98) | (1.20) | (1.11) | (0.96) |
| Personal-outgroup divergence | | | 0.24 | | |
| | | | (1.20) | | |
| Intergroup divergence | | | | 0.39 | |
| | | | | (1.11) | |
| Self-reported imagined otherness | | | | | 8.77*** |
| | | | | | (0.87) |
| Democrat | | 8.43*** | 8.43*** | 8.35*** | 9.36*** |
| | | (2.20) | (2.20) | (2.21) | (2.08) |
| Strong partisan | | 13.20*** | 13.20*** | 13.19*** | 8.94*** |
| | | (2.32) | (2.32) | (2.32) | (2.23) |
| Extreme conservative | | 7.76* | 7.76* | 7.80* | 6.68* |
| | | (4.04) | (4.05) | (4.05) | (3.81) |
| Ideological extremity | | 2.30 | 2.26 | 2.25 | 1.55 |
| | | (1.44) | (1.45) | (1.45) | (1.35) |
| Constant | 19.07*** | 2.36 | 2.44 | 2.50 | −23.96*** |
| | (1.03) | (2.43) | (2.46) | (2.46) | (3.47) |
| N | 772 | 772 | 772 | 772 | 769 |
| df | 770 | 766 | 765 | 765 | 762 |
| R² | 0.034 | 0.156 | 0.156 | 0.156 | 0.256 |

Standard errors in parentheses *p < 0.05, **p < 0.01, ***p < 0.001 95% CIs for "Imagined otherness": [3.38, 7.42], [3.46, 7.30], [2.90, 7.59], [3.01, 7.37], [0.76, 4.54] Exact P-values for "Imagined otherness": 0.000, 0.000, 0.000, 0.000, 0.006.

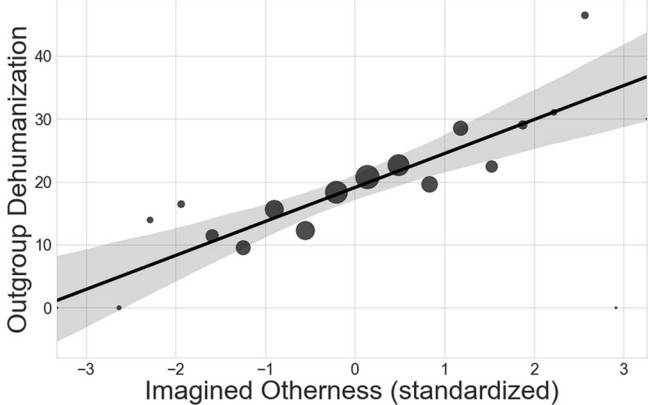

**Fig. 5 | Imagined otherness and outgroup dehumanization.** Estimated bivariate relationship between imagined otherness and outgroup dehumanization via an OLS regression. Gray bands represent 95% confidence intervals estimated using bootstrapping (1000 repetitions). Each dot shows the mean value of outgroup dehumanization exhibited by participants with that level of imagined otherness. Dots are sized by number of participants with that level of imagined otherness. N = 771.

more closely aligned their ingroup schema. We thank one of the reviewers for suggesting this supplemental analysis.

In the second analysis, we unpack how the relationship between generalized outgroup schema distance and outgroup dehumanization varies

## a

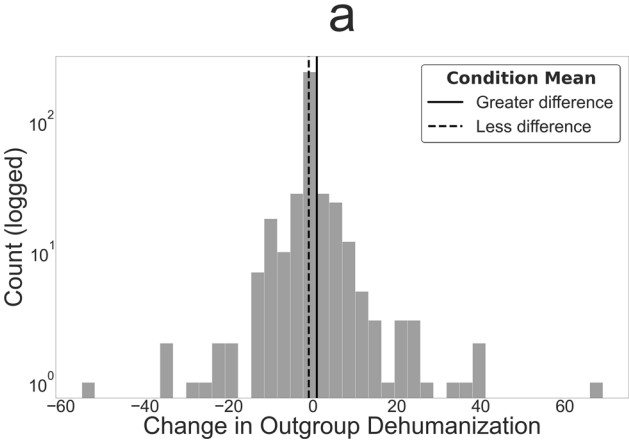

## b

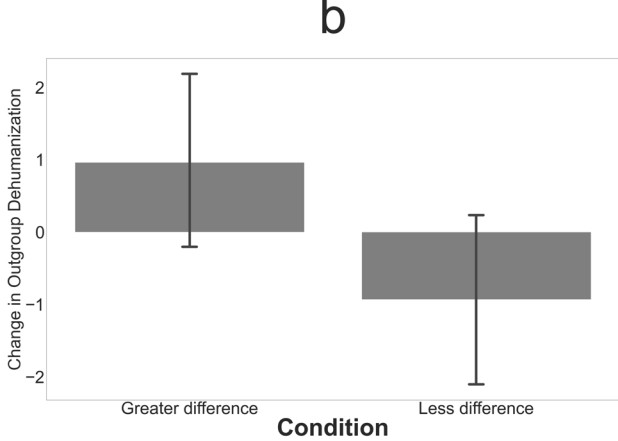

**Fig. 6 | Experimental results: imagined otherness and outgroup dehumanization.** **a** Histogram of change in outgroup dehumanization. Mean of greater difference condition is plotted with a solid line; mean of less difference condition is plotted with a dashed line. **b** Means of low and greater difference conditions with 95% confidence intervals estimated via bootstrapping (1k repetitions). In the greater difference condition, $N = 197$. In the less difference condition, $N = 201$.

depending on the *content* of schemas: positive, negative, or neutral. Specifically, we calculated separate measures of outgroup schema distance based on positive, negative, and neutral terms. Counter to our expectations, we only found statistically significant evidence that—in a fully saturated model—outgroup schema distance based on negative terms is positively related to outgroup dehumanization. As this analysis illustrates, one key advantage of our schema elicitation task over a simple self-report of perceived difference between an outgroup and most people is that our measurement strategy allows researchers to unpack the content of schemas that matter most for dehumanization. We conjecture that the relative importance of positive versus negative versus neutral terms will vary based on the nature of the focal concept under study (e.g., uniformly acclaimed versus commonly stigmatized category).

### Experimental study

Our manipulation check indicated that our experimental condition successfully changed participants' levels of the self-report of imagined otherness. Specifically, participants in the greater difference condition increased their self-report of perceived difference (i.e., after seeing the manipulation) than did participants in the less difference condition according to both a two-tailed independent samples t-test ($t = 2.91$, $d = 0.29$, 95% $CI_{\mu1-\mu2} = [0.08, 0.42]$, $df = 396$, $p = 0.004$), as well as a two-tailed Wilcoxon rank-sum test ($z = 2.12$, $CI_z$ [bootstrapped, 1k repetitions] $= [0.48, 3.75]$, $p = 0.034$). The effect was, in absolute terms, small: participants in the less difference condition reported a decrease in perceived difference of 0.05 points (with a post-treatment mean of 3.49), while participants in the greater difference condition reported an increase of 0.20 points (with a post-treatment mean of 3.66).

Finally, participants in the greater difference condition increased their outgroup dehumanization more than did participants in the less difference condition according to both a two-tailed independent samples t-test ($t = 2.19$, $d = 0.22$, 95% $CI_{\mu1-\mu2} = [0.23, 3.63]$, $df = 396$, $p = 0.029$) and a two-tailed Wilcoxon rank-sum test ($z = 2.68$, $CI_z$ [bootstrapped, 1k repetitions] $= [0.90, 4.46]$, $p = 0.007$). Figure 6 plots the overall distribution of our dependent variable and shows the differences in means across conditions. It is worth noting that the effect size was relatively small, with differences in means of less than 2 points. This suggests that, although our manipulations did yield significant differences and these results establish a causal link between perceived schematic difference and outgroup dehumanization, stronger manipulations (e.g., repeated exposures and information from more legitimate sources) may be needed to arouse levels of dehumanization that would lead to actual negative behavior exhibited toward outgroup members.

### Discussion

Why do people deny the humanity of other groups? A prominent preoccupation in the postwar era, sociological inquiry of dehumanization appears to have waned since. Recent work exploring blatant dehumanization from a sociological perspective has predominantly focused on institutional processes of organized violence, state propaganda, and demonizing public discourse. Social psychological research has instead focused on blatant and subtle dehumanization arising from the denial of mind to others[29]. Integrating insights from social psychology and cultural sociology, we offer a theoretical account of blatant dehumanization's subtle, cognitive origins—one in which it arises not from the denial of mind to others but from active consideration of their minds.

### Dehumanization's origins and the representation of social boundaries

Drawing on research in cultural sociology on shared understandings that manifest in the form of schemas [e.g., ref. 33], we specifically trace its origins to imagined otherness—when people perceive members of other groups as thinking about consequential aspects of the social world in fundamentally different ways than most people do. Focusing on the context of U.S. politics and the contested concept, America, we first demonstrate that imagined otherness is positively related to the tendency to blatantly dehumanize members of the opposing political party. Our second study uses an experimental design to pin down the causal link between perceived schematic difference and blatant dehumanization. Together, the studies demonstrate that as perceptions of schematic distance between a typical outgroup member and a typical person grow, the outgroup's perceived humanity declines.

Although our project focused on the U.S. political context, our findings have broader implications for the study of social boundaries between groups. As studies of categorical inequality have demonstrated, perceptions of inherent differences between groups relate to disparities along a variety of identity boundaries, from the essentialized beliefs at the root of gendered occupational segregation[96] to differences in attributions of immigrant illegality by country of origin[97]. We contribute to these and related literatures by highlighting perceived schematic difference as an important mechanism through which group boundaries are cognitively represented. Sociological work on boundaries has tended to focus on their manifestations either in patterns of intergroup contact [e.g., ref. 98] or symbolic distinctions [e.g., ref. 99]. These studies implicitly assume that individuals internalize the patterns they observe as abstract perceptions of difference [e.g., ref. 48]. Individuals might find it difficult, or socially undesirable, to articulate the reasons for their perceptions of intergroup difference. Thus, as our study

demonstrates, there is value to unpacking general perceptions of difference into specific schematic associations. Indeed, understanding *how* group difference is perceived in the form of distances in schematic space helps account for who is likely to engage in blatant dehumanization of an outgroup above and beyond social identity-based mechanisms and abstract notions of group difference.

## Political polarization in the United States

Increasing political polarization in the U.S. has puzzled scholars of American politics. While interparty hostility has been unequivocally on the rise, ordinary Americans have not exhibited an equally clear and systematic increase in ideological and policy polarization[73,79]. A common explanation for this phenomenon is partisan sorting: the increased alignment between voters' party affiliations and their ideological and policy preferences[74,100]. Even if Americans have, overall, not become more polarized in their opinions, proponents of this view contend that greater congruence between voters' ideology and partisanship likely increases their perceptions of a social boundary between these two political groups.

Our results suggest that partisan sorting is, at best, a partial explanation for the rise of polarization. We find that a particularly potent manifestation of polarization—blatant dehumanization of the opposing political party—is associated with perceived schematic difference above and beyond the strength of party identity and ideological extremism. Imagined otherness thus appears to represent a complementary mechanism through which political polarization occurs. We anticipate that exploring heterogeneity in imagined otherness across geographies and over time might help to account for the recent steep rise in polarization in the U.S.

Although our results do not directly speak to outcomes beyond outgroup dehumanization, we conjecture that they may have implications for other attitudes and behaviors at the heart of political polarization. For example, a puzzling finding in the U.S. context is that outparty animosity does not seem to be as closely linked to support for anti-democratic strategies and candidates as the literature had largely assumed[101]. Given research on how dehumanizing a group can legitimize otherwise unacceptable behavior[30], our findings point to a potential answer to this puzzle. It may be that political dehumanization, partially caused by imagined otherness, moderates the relationship between affective polarization and support for anti-democratic practices. In other words, partisans who are highly polarized *and* who exhibit outgroup dehumanization may be the ones most likely to support anti-democratic practices and candidates. We leave an empirical examination of this relationship to future work.

## Measuring meaning

Sociologists of culture have long been interested in measuring meaning structures[102]. Recent years have seen a proliferation of methods aimed at identifying the semantic structures of association that make up people's schematic cognition. As we outlined above, different methods have different advantages and disadvantages. Yet existing methods are designed, without exception, to tap personal schemas. Addressing this limitation, we used an approach—building on and extending existing methods—to elicit both personal and attributed schemas. Although we do not claim that our approach is unambiguously superior to previous ones and recognize that every method has limitations, we believe the measurement strategy we deploy has two main advantages relative to prevailing methods.

First, because our technique relies on associations with similarly valenced words, it avoids confounding affect with interpretation. Our approach is not, however, without limitations. For example, asking people of faith which negative words are associated with the concept God might lead some participants to experience strong discomfort. More broadly, our technique is sensitive to the fixed set of associated concepts presented to respondents and predetermined by the researchers. As we did for this study, researchers using it would benefit from generating and pre-testing a broader set of words that are associated in participants' minds with their focal concept of interest and from using word embedding models to guide their choices of which words to ultimately include in the task. More work is also needed to understand which focal words—e.g., America—are most relevant for which attitudes and behaviors and the robustness of the relationship between perceived schematic distance and outcomes of interest when using different plausible focal words–e.g., Freedom or Democracy.

Second, our technique is based on a simple and clearly delineated association task. This simplicity makes the tool scalable and easy to apply at different schematic levels without significantly increasing respondent cognitive load or risking participant attrition. Moreover, the straightforward nature of the task makes it easy to compare associations between and within respondents. In contrast, consider the complex association task that Hunzaker and Valentino[33] use, wherein respondents are asked to compare many pairs of concepts. Measuring the similarity between respondents' resulting matrices of association is challenging, requiring maximum-likelihood estimations that rely on various assumptions. Using this approach also necessitates making strong a priori assumptions about how the population is divided into groups with respect to their personal schemas. Hunzaker and Valentino[33], for instance, perform their analysis based on the assumption that Republicans and Democrats have different personal schemas of poverty. While this seems reasonable and is consistent with the analyses we report in the Supplementary Note 2, it is not axiomatic and requires validation. Moreover, as the authors' results show, it is not clear that party identities explain differences in the personal schema of poverty better than other group boundaries. With our measurement strategy, evaluating whether two groups are statistically different in their personal schemas requires just a simple statistical test.

The efficiency of our measurement strategy also makes it easy to extend the approach to the broader study of schematic distances and intergroup attitudes and behavior. For example, the validation checks presented in supplementary note 2 demonstrate that the perceived distance between one's ingroup schema and one's personal schema is positively related to a different outcome of interest—affective polarization. With minor adjustments, this measurement technique can be configured to reveal other causes and consequences of perceived schematic distance. For example, consider prevalent conceptions about men being from Mars and women from Venus. Popularized by John Gray[103], this interstellar metaphor alludes to men's and women's presumably inherently different ways of understanding romantic relationships. A schema elicitation task centered on the word love, for example, might evaluate the extent to which men and women actually think about love differently. Opposite-gender schemas measured using this approach can evaluate the extent to which the group perceptions by members of each gender exaggerate or underestimate these differences, as well as whether they correctly evaluate the extent to which opposite-gender schemas are homogeneous or varied.

While the present study focused on static perceptions of difference, we also see great potential in the use of this tool in longitudinal study designs that involve between-person comparisons, within-person comparison, and the interaction between the two. We conjecture, for example, that newcomers to a group will tend to have schemas that diverge from those of typical group members but that this between-person distance will decline for newcomers who effectively socialize into the group. Similarly, individuals who perceive greater schematic distance between their ingroup and outgroup might over time have personal schemas that migrate closer to their ingroup schema, perhaps leading them to identify more strongly with their ingroup. Finally, newcomers who effectively socialize into a group and have schemas that begin to resemble those of typical group members might similarly experience a corresponding decrease in their personal-ingroup schematic distance. In general, we suspect that time-varying measures of perceived schematic difference may prove to be instrumental in studies of intergroup dynamics.

## Limitations

This study has several limitations, which point to avenues for future research. First, our theory posits that imagined otherness centers around understandings of important or consequential facets of the social world. This implies that our observed results would be weaker had we measured

https://doi.org/10.1038/s44271-024-00087-4 **Article**

understandings of such ordinary concepts as chairs or hamsters. We do not, however, provide any evidence here to support such a conjecture. We leave to future work the task of identifying how important or consequential a focal concept must be for the mechanism of imagined otherness to be operative. Similarly, we do not test the scope conditions of our theory—for example, the extent to which it extends to social groups for which the prototype of most people does not serve as a reference point for legitimacy.

Next, although we provide strong evidence for a causal effect of imagined otherness on outgroup dehumanization, our theory is not incompatible with a reverse causal pathway—namely, that outgroup dehumanization begets imagined otherness. We leave to future work the tricky research design task of experimentally manipulating outgroup dehumanization. In a similar vein, our experimental paradigm could be extended to causally identify the correlational evidence we report that personal-ingroup divergence moderates the relationship between imagined otherness and outgroup dehumanization. Finally, recognizing the limitations of the Ascent of Man scale, future research could examine other how imagined otherness relates to other, more subtle measures of dehumanization, such as infrahumanization[24].

## Conclusion

Given the cruelties that humans are capable of inflicting upon others, understanding the origins of blatant dehumanization is of paramount importance if we aim to ultimately curb such atrocities. This study demonstrates that merely perceiving the other as seeing the world in a fundamentally different way can lead people to conceive of them as subhuman. Thus, finding ways to help others imagine oneness, rather than otherness, with outgroup members may prove critical in the quest to bridge the bitter divisions that exist in society.

## Data availability

The data necessary to replicate all results in this paper can be found at the first author's GitHub: https://github.com/AustinVL/ImaginedOtherness/.

## Code availability

The analysis code necessary to replicate all results in this paper can be found at the first author's GitHub: https://github.com/AustinVL/ImaginedOtherness/.

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

## Acknowledgements

The authors have no funding sources to disclose. We would, however, like to thank the following groups for their invaluable feedback on this work: the University of Michigan Interdisciplinary Committee on Organizational Studies Seminar, the University of Maryland Smith School of Business Cross-Disciplinary Seminar Series, The Columbia Business School Management Division Seminar, The Yale School of Management Organizational Behavior Seminar

## Author contributions

Av.L., A.G. and S.B.S. jointly developed the theory and research design. Av.L. ran the study and analyzed the results. Av.L., A.G. and S.B.S. jointly wrote the manuscript.

## Competing interests
The authors declare no competing interests.
