## [Peer Review File · Communications Psychology]

31st Aug 23

Dear Dr. van Loon,

Thank you for your patience during the peer-review process. Your manuscript titled "Imagined Otherness: Perceived Schematic Difference Can Fuel Dehumanization" has now been seen by 3 reviewers, and I include their comments at the end of this message. They find your work of interest, but raised some important points. We are interested in the possibility of publishing your study in *Communications Psychology*, but would like to consider your responses to these concerns and assess a revised manuscript before we make a final decision on publication.

We therefore invite you to revise and resubmit your manuscript, along with a point-by-point response to the reviewers. Please highlight all changes in the manuscript text file.

Editorially, we consider it important that the revised manuscript provides an improved literature review and better conceptualization of the key constructs. This is particularly important to broaden the impact of a piece that sits in a research area of multidisciplinary interest. Reviewers 1 and 3 express concerns regarding some methodological choices which should be addressed or, where not possible, discussed as limitations. While we agree with the reviewers that the utility and limitations of the new task should be discussed at greater length, we do not encourage you to remove this from the manuscript (neither should any other preregistered components be removed). The reviewers also request additional analyses, which should be performed. Please include details regarding participant consent in the methods section.

Please use the following link to submit your revised manuscript, point-by-point response to the referees' comments (which should be in a separate document to any cover letter) and the completed checklist:

[link redacted]

Please do not hesitate to contact me if you have any questions or would like to discuss these revisions further. We look forward to seeing the revised manuscript and thank you for the

opportunity to review your work.

Best regards,

Jennifer Bellingtier

Jennifer Bellingtier, PhD
Senior Editor
Communications Psychology

EDITORIAL POLICIES AND FORMATTING

Editorial Policy: Policy requirements (Download the link to your computer as a PDF.)

Furthermore, please align your manuscript with our format requirements, which are summarized on the following checklist:

Communications Psychology formatting checklist

and also in our style and formatting guide Communications Psychology formatting guide .

* **CODE AVAILABILITY:** All Communications Psychology manuscripts must include a section titled "Code Availability" at the end of the methods section. In the event of publication, we require that the custom analysis code supporting your conclusions is made available in a publicly accessible repository; at publication, we ask you to choose a repository that provides a DOI for the code; the link to the repository and the DOI will need to be included in the Code Availability statement. Publication as Supplementary Information will not suffice. We ask you to prepare code at this stage, to avoid delays later on in the process.

* **DATA AVAILABILITY:**

All Communications Psychology manuscripts must include a section titled "Data Availability" at the end of the Methods section or main text (if no Methods). More information on this policy, is available at <http://www.nature.com/authors/policies/data/data-availability-statements-data-citations.pdf>.

At a minimum the Data availability statement must explain how the data can be obtained and whether there are any restrictions on data sharing. Communications Psychology strongly endorses open sharing of data. If you do make your data openly available, please include in the statement:

We recommend submitting the data to discipline-specific, community-recognized repositories, where possible and a list of recommended repositories is provided at

<http://www.nature.com/sdata/policies/repositories>.

If a community resource is unavailable, data can be submitted to generalist repositories such as figshare or Dryad Digital Repository. Please provide a unique identifier for the data (for example a DOI or a permanent URL) in the data availability statement, if possible. If the repository does not provide identifiers, we encourage authors to supply the search terms that will return the data. For data that have been obtained from publicly available sources, please provide a URL and the specific data product name in the data availability statement. Data with a DOI should be further cited in the methods reference section.

REVIEWERS' EXPERTISE:

Reviewer #1 dehumanization, sociology

Reviewer #2 dehumanization, group dynamics

Reviewer #3 dehumanization, social cognition

REVIEWERS' COMMENTS:

Reviewer #1 (Remarks to the Author):

Building on past work in sociology and psychology, this innovative study develops a new theory of dehumanization, focused on the perceived distance between the cultural schemas of a focal outgroup and the generalized other (i.e., "most people"). The theory predicts that the greater the perception of the outgroup's schematic distance from that of the generalized other, the more likely that outgroup is to be dehumanized. The author applies this model to the concept of "America" and the dehumanization of partisan outgroups (i.e., of Democrats among Republican respondents and Republicans among Democratic respondents). To test the theory, the study employs a novel schema elicitation task that helps overcome the limitations of explicit and implicit approaches used in past research. The (preregistered) results are in line with the theory's predictions.

I found the paper's arguments compelling, the research design rigorous, and the results interesting. In my view, the paper makes important contributions to the scholarly understanding of dehumanization processes, affective polarization, and cultural measurement. I do have some comments and questions, but these largely call for the clarification and elaboration of existing claims rather than for major revisions to the study as a whole.

1. The author distinguishes the proposed theory from previous scholarship that predicates dehumanization on the denial of “mind” to members of the outgroup. This seems reasonable, but it would be helpful to elaborate and clarify this discussion. What exactly does “mind denial” involve (how do scholars define “mind” and how do they know that respondents perceive outgroups as not possessing full minds)? Moreover, is the distinction between the mind denial mechanism and the proposed schema incongruity mechanism as sharp as the author suggests? Isn’t it the case that schema incongruity often involves stereotypical perceptions of the outgroup’s stupidity, ignorance, and/or moral shortcomings and therefore implies outgroup members’ diminished quality of mind?

2. I find the study’s central claim that dehumanization stems from the incongruity between the schemas of outgroups and the generalized other to be convincing. I do wonder, however, about boundary cases where this mechanism may not hold. In particular, it is likely that certain morally self-righteous minorities (e.g., orthodox religious groups, political extremists) subscribe to the belief that most people—and not just specific outgroups—are both morally deficient and wrong in their views of the world. If so, members of these ideologically extreme groups may not perceive the generalized other as a legitimate point of reference for evaluating schematic meanings. It’s also possible that they will dehumanize most people who do not share their group’s perceptions of the world. If so, the hypothesis about imagined otherness being rooted in perceived distance from the generalized other may not hold for these groups. Does the author agree that this alternative scenario poses a plausible departure from the proposed theoretical model? And if so, could the paper offer a more explicit statement of the theory’s scope conditions, including contexts under which its assumptions may not hold?

3. The author argues that dehumanization does not necessarily imply outgroup antipathy. But is this really true? And is this claim necessary for the theory? Isn’t it the case that dehumanization generates feelings of disgust, which are in turn associated with negative affect toward an outgroup? Certainly, not all outgroup antipathy involves dehumanization, but it’s difficult for me to imagine dehumanization without outgroup antipathy.

4. The research design is clever and well-suited to the study’s objectives. That said, I found the decision to subject respondents to forced-choice selections from sets of negative and positive attributes of the focal concept to be potentially problematic. What if the ingroup’s understanding of the given concept is predicated on only positive or only negative attributions? How do we expect respondents to meaningfully choose attributes of the opposite valence? Might they not only see such attributes as inapplicable but also perceive their forced selection as an affront to their identities? How might the resulting measurements be affected by respondents’ experience of discomfort during the elicitation task? The paper touches on this in the appendix, but only briefly. I would bring this into the main text and elaborate it. I also found the discussion of the distinction between attributes respondents associate with the focal concept vs. their feelings about the concept (as a weakness of past methods that the current approach seeks to overcome) to be rather unclear. It would be helpful to explain this further.

5. The author excludes all respondents who completed the elicitation task in less than half the median response time. Given the mixed recommendations in the methodological literature about excluding survey speeders from experimental samples, it would be helpful to know how sensitive the results are to this decision. Would retaining all respondents or those who completed the survey in more than, say, 25% of the median time alter the findings?

6. The paper makes several comparisons between the types of schemas measured by the survey instrument, but the one comparison that is given the least attention in the main text is the relative congruence between personal and ingroup schemas. This is attended to in the appendix as a robustness check (those with stronger ingroup attachment exhibit greater congruence), but could this measure be used directly in the research design as well? My thinking was that personal-vs-ingroup schema congruence could be an interesting alternative measure of ingroup attachment to self-reported partisan identification (or ingroup identity in general). It could also serve as a theoretically interesting moderator of the link between imagined otherness and dehumanization. Leaving it to the supplemental robustness check section seems like a missed opportunity.

7. The elicitation task introduced by the study seems like an effective measurement strategy for identifying the perceived schematic otherness of outgroup members. At the same time, the study also includes explicit self-reported measures of schematic otherness and uses them as a control in one of the regression models. Interestingly, this explicit measure is modestly correlated with the measure generated by the elicitation task, but it is even more highly correlated with dehumanization than the elicitation task measure. It would be helpful for the author to be clearer about the value added by the (clever but time-consuming) elicitation task above and beyond the self-reported measure. Why not just use the latter given its good performance in the models? What insights can be gleaned from the elicitation task that cannot be gleaned from the self-reported measure?

8. Dehumanization is measured using the Ascent of Man scale, as had been the case in prior psychological studies. I would have liked to see a fuller discussion of this instrument's past validation, its advantages over alternative measures, and some of its limitations. I worry in particular that the images used in the scale are blunt (essentially asking respondents to compare specific groups to apes), evoke evolutionary theory (which some respondents may reject wholesale), and are inherently gendered (only males are depicted in the prompt). How might these characteristics of the instrument affect the measurement of dehumanization in the study? Did the author consider alternative measurement strategies?

9. A few claims in the paper could use additional citations. In particular, the sociological research on dehumanization by Aliza Luft seems conspicuously missing from the references (and calls into question the statement that "sociological inquiry of dehumanization appears to have waned since [the post-war years]"). The theory section should engage in particular with Luft's findings that dehumanization often follows from genocidal violence rather than preceding it. Another literature that's frequently referenced in the paper but rarely actually cited is the work on affective polarization by scholars like Shanto Iyengar, Jamie Druckman, Yphtach Lelkes, Matthew Levendusky, and Lily Mason. Finally, for recent work on partisan differences in the conceptualization of America, see Sides, Tesler, and Vavreck's 2018 book *Identity Crisis* and my 2021 AJS article, "The Partisan Sorting of 'America'" (no pressure to cite the latter, of course, but it may offer some helpful references and findings to further justify the present test case).

It was a pleasure to read this terrific piece of research. I look forward to recommending it to others and citing it myself in the future.

Bart Bonikowski
New York University

Reviewer #2 (Remarks to the Author):

Overall I found this an interesting and very well written article that seeks to establish a new understanding of the emergence of blatant dehumanization that sets it apart from existing scholarship focusing either on institutional determinants or dehumanization as a denial of mind of others. Instead, it is proposed, dehumanization may also occur via the process of imagined otherness in which members of a group perceive a discrepancy between their outgroup schema and their generalized other schema. This is then demonstrated with reference to two studies. The article then offers a discussion on the results and the wider consequences of its findings.

The article overall succeeds in demonstrating its working hypothesis and uses a relevant case to demonstrate the importance of imagined otherness in increasing a tendency towards dehumanization. There are, however, a number of points I would encourage the author(s) to clarify and substantiate:

1) While there is an attempt to differentiate the main argument proposed in the article from existing research, the article lacks a proper literature review of the positions it sets itself apart from. There are some sections in which some of the relevant literature in sociology (though this seems to be a catch-phrase also including research on dehumanization in political science, political theory and philosophy) and (social) psychology is introduced though the article lacks a proper critical engagement with this literature to give the reader a sense of what the different positions are, how they are related to each other and how the current research sets itself apart from these existing positions. Such an outline would also benefit from a wider acknowledgement of dehumanization research and varying concepts of dehumanization (beyond institutional and cognitive accounts there are prominent positions which see dehumanization as a denial of moral status, as a discursive strategy or understand it behaviorally). While the article cannot offer a comprehensive review, it would benefit from a more balanced and substantial consideration of the existing literature, not least as it would further emphasize the novelty of the approach offered here.

2) The article lacks a clear delineation of the concept of dehumanization. In the accounts it highlights at the beginning we find a clear sense of what dehumanization consists in (denial of mind, animalization, mechanization etc), i.e. these accounts establish clearly what they take to be the central component of 'humanity' that is being denied in cases of dehumanization. The article, however, does not offer such a conceptual delineation, so the reader is left wondering what dehumanization actually consists in and what element(s) of being human are actually denied in cases of 'imagined otherness'. At some point it seems that dehumanization describes the distance between a person's outgroup schema and the generalized other schema but then the article says that 'the tendency to dehumanize increases with the distance between these schemas' which implies that the discrepancy is distinct from dehumanization, so we still don't know what is meant by 'dehumanization'.

Further along the article uses a slider scale linked to the 'ascent of man' model; even if this would be the way in which the article perceives dehumanization (ranking on a scale from ape to fully upright humans), the problem is that this scale was given to respondents as the only way to rate democrats and republicans – i.e. respondents did not choose to describe their perceptions of these groups on a ape to human scale. Whether their ranking actually expresses a belief that democrats/republicans are more ape-like (and hence less human) is unclear (we only know they rank them 'lower' but since the ascent of man ladder was the only way to express this, we cannot conclude that respondents substantially agreed with or would subscribe to expressing their ranking in relation to an ape-human

categorization). Overall, the article needs to be much clearer as to the factor(s) central to the concept of 'human' that are being denied in order to justify using the concept of 'dehumanization' to describe the process it identifies.

3) Linked to the points above, it is also somewhat unclear what the connection between imagined otherness and dehumanization is. The article early on states that imagined otherness CAN arouse blatant dehumanization (page 2) with imagined otherness meaning "that a prototypical member of a social group construes an important facet of the social world in ways that diverge from the way most humans understand it". It remains however, somewhat unclear, a) under what conditions imagined otherness DOES arouse blatant dehumanization (the stipulation that it CAN do so implies that it is not invariably the case) and b) what the 'important facets' are (it seems to imply that a group can construe (non-important) facets of the social world that do diverge from the way most humans understand them without leading to the process described here). The question then would be not only what these important facets are but also who determines them.

4) Finally, while the article provides some reflections on the consequences of its research specifically in terms of political polarization in the US, it remains somewhat vague as to the extent blatant dehumanization is linked with (political) behavioural outcomes, an aspect that much of dehumanization research is focused on (e.g. is the presence of dehumanization in US political discourse linked to increasingly extreme forms of political activism?). I am not sure if the article can or does want to make any claims in this respect but it would be interesting to know whether this research can further contribute to the link between dehumanizing cognitive dispositions and behavioural outcomes (especially since this link has come under increasing criticism in recent studies – e.g. Harriet Over, Johannes Lang, Nicolas Mariot).

Overall, I think this article presents an interesting and novel contribution to existing research on dehumanization though it would benefit from a more critical and substantive engagement with the existing literature, a clearer conceptual delineation of what is meant by dehumanization and a more explicit discussion of the link between imagined otherness and the occurrence of dehumanization.

Reviewer #3 (Remarks to the Author):

The authors reported two studies, one correlational and one experimental, that examined the association between "imagined otherness" and blatant dehumanization of outgroup. Specifically, they hypothesized and confirmed that the larger the perceived differences in the concepts related to the focal word "America" between an outgroup (i.e., members of an opposing political party) and a typical human being, the greater this outgroup would be dehumanized.

I found the idea very intriguing, and agree that we might dehumanize social targets who see the world in strikingly different ways than we, or people typically do. We may see them as less sophisticated, or are deficient in feeling or depth. This said, I don't think the manuscript is ready for publication in the current form.

First, the introduction seems a little scattered. Why would "imagined otherness" elicit blatant dehumanization? Is it because that imagined otherness is linked with a "diminished mind", or because it implies a distinct, and "less than human" essence? The authors talked about both accounts but failed to provide a clear explanation.

A related question is, are all the social perceptions people have a core element of mind or human essence? The authors used the specific concept of "America" as the focal word, which I think, may out of reasons not (but should be) explicitly stated. Can the results be generalized to other concepts such as "democracy", "country", "freedom", "love"...? Or, do the concepts we should use in the test depend on the particular outgroup we are evaluating? This is a vital question the authors should think about because it is not only closely tied to the robustness and generalizability of the results, but also of crucial theoretical importance.

The authors defined "imagined otherness" as the outgroup-generalized other schematic difference, which does make sense. But how about the personal-outgroup and ingroup-outgroup differences? Are they similarly associated with dehumanization? After all, people often see themselves and ingroup as more human than others. I think it doesn't hurt to also report these results.

More demographic information about the participants may be given.

The authors reported a more straightforward measure of perceived otherness, which they named self-reported othering. By asking people how much they believe an outgroup's social perceptions are different from the perceptions of generalized others, we can know directly how aberrant the outgroup's thinking is thought to be. I noticed that this index showed a greater association with blatant dehumanization than the measure of imagined otherness. Then why bother devising the schema elicitation task? Although the task seems more objective than the subjective report, it does have many limitations. First, we need to choose an appropriate focal word (Or does any word work? I'm afraid not..). Second, it seems very laborious and bothersome to generate the many related concepts. Third, we have to force the respondents to choose from the predetermined concepts. Subjectivity may be introduced in all the steps. Instead, with the subjective report, we may have a more comprehensive understanding of perceived otherness by asking participants how large are the differences in perceiving concepts such as "human", "America", "country", "freedom", "morality", "love"...by a certain outgroup and by a typical person! In a word, I do have serious concerns about the measure.

I know it's beyond the scope of this research but do wonder if the reversed causal link also holds. If we dehumanize an outgroup more, would we see a larger schematic distance between them and ourselves? It seems to make sense too.

Although the studies focus on blatant dehumanization, the authors may also measure and report subtle dehumanization to examine whether imagined otherness uniquely predicts blatant dehumanization.

In sum, I really like the idea the authors proposed. Yet, I don't think they have made it clear why imagined otherness leads to blatant dehumanization, or whether the measure of imagined otherness is valid or necessary.

RESPONSE TO REVIEWER 1

1. *The author distinguishes the proposed theory from previous scholarship that predicates dehumanization on the denial of “mind” to members of the outgroup. This seems reasonable, but it would be helpful to elaborate and clarify this discussion. What exactly does “mind denial” involve (how do scholars define “mind” and how do they know that respondents perceive outgroups as not possessing full minds)? Moreover, is the distinction between the mind denial mechanism and the proposed schema incongruity mechanism as sharp as the author suggests? Isn’t it the case that schema incongruity often involves stereotypical perceptions of the outgroup’s stupidity, ignorance, and/or moral shortcomings and therefore implies outgroup members’ diminished quality of mind?*

Thank you for pushing us to clarify our theoretical argument and contribution. On page 4, we’ve extended our discussion of the denial-of-mind mechanism and how imagined otherness differs from it. We hope this more clearly differentiates the process of imagined otherness from mind denial.

2. *I find the study’s central claim that dehumanization stems from the incongruity between the schemas of outgroups and the generalized other to be convincing. I do wonder, however, about boundary cases where this mechanism may not hold. In particular, it is likely that certain morally self-righteous minorities (e.g., orthodox religious groups, political extremists) subscribe to the belief that most people—and not just specific outgroups—are both morally deficient and wrong in their views of the world. If so, members of these ideologically extreme groups may not perceive the generalized other as a legitimate point of reference for evaluating schematic meanings. It’s also possible that they will dehumanize most people who do not share their group’s perceptions of the world. If so, the hypothesis about imagined otherness being rooted in perceived distance from the generalized other may not hold for these groups. Does the author agree that this alternative scenario poses a plausible departure from the proposed theoretical model? And if so, could the paper offer a more explicit statement of the theory’s scope conditions, including contexts under which its assumptions may not hold?*

We completely agree that scope conditions are an important but currently missing element of our theoretical formulation. We now offer some speculative thoughts on page 5 and call for future research to explore this empirically on page 25.

3. *The author argues that dehumanization does not necessarily imply outgroup antipathy. But is this really true? And is this claim necessary for the theory? Isn’t it the case that dehumanization generates feelings of disgust, which are in turn associated with negative affect toward an outgroup? Certainly, not all outgroup antipathy involves dehumanization, but it’s difficult for me to imagine dehumanization without outgroup antipathy.*

Our reading of the social psychological literature on dehumanization suggests that dehumanization (and especially blatant dehumanization) does indeed have effects on outcomes that are distinct from antipathy. Indeed, we concur with the overwhelming consensus in this literature—that dehumanization and dislike are analytically distinct. Consider, for example, that people often have very positive sentiments towards non-human entities such as pets.

However, we concede that these constructs are correlated in our empirical setting and that we need not make strong claims about their relationship to develop our theory. We agree that we could do a better job of explicating these points in the manuscript, and have added some additional discussion of this important analytical issue throughout the manuscript, but especially in footnote 4.

4. The research design is clever and well-suited to the study’s objectives. That said, I found the decision to subject respondents to forced-choice selections from sets of negative and positive attributes of the focal concept to be potentially problematic. What if the ingroup’s understanding of the given concept is predicated on only positive or only negative attributions? How do we expect respondents to meaningfully choose attributes of the opposite valence? Might they not only see such attributes as inapplicable but also perceive their forced selection as an affront to their identities? How might the resulting measurements be affected by respondents’ experience of discomfort during the elicitation task? The paper touches on this in the appendix, but only briefly. I would bring this into the main text and elaborate it. I also found the discussion of the distinction between attributes respondents associate with the focal concept vs. their feelings about the concept (as a weakness of past methods that the current approach seeks to overcome) to be rather unclear. It would be helpful to explain this further.

We thank the reviewer (and the other reviewers) for pushing us to think about these potential limitations of our measurement strategy. We have three main responses.

First, in footnote 5 we now acknowledge that, when measuring schemas related to concepts for which respondents’ interpretations are overwhelmingly positive or negative, the approach we take here might need to be modified. That said, given that our sample includes Republicans and Democrats, many of whom might be patriotic but nevertheless see the challenges that America faces, we suspect that few of our participants hold unambiguously positive or negative views of the country.

Second, to more directly rule out the possibility that our results are an artifact of people being forced to categorize terms that are odds with how they actually feel about America, we perform an exploratory (i.e., not pre-registered) analysis in which we replicate Table 2 from the manuscript using an alternative measure of imagined otherness—one defined solely using word choices made among neutral concept sets (see Table R&R1 below). Given that we obtain comparable results in this analysis, we hope this analysis assuages your concerns about the potentially confounding effects of participant discomfort in completing the categorization task.

Table R&R1. Outgroup Dehumanization, Imagined Othering, and PID

	M1	M2	M3	M4	M5
Imagined Othering (Neutral only)	4.61*** (1.04)	5.46*** (0.99)	5.37*** (1.17)	4.48*** (1.17)	3.72*** (0.94)
Personal-outgroup Divergence			✓		

Intergroup Divergence				✓	
Self-reported Imagined Othering					✓
Democrat		✓	✓	✓	✓
Strong ID		✓	✓	✓	✓
Ideological Extremity		✓	✓	✓	✓
Constant	19.1	8.6	8.7	9.1	5.6
R^2	0.03	0.15	0.15	0.15	0.26
N	771	771	771	771	771

Imagined othering, POD, intergroup divergence, and Outgroup Dehumanization are all standardized.
Standard errors in parentheses (excluded for constants)

* $p < 0.05$, ** $p < 0.01$, *** $p < 0.001$ (excluded for constants)

Finally, we clarify in our discussion on pages 10-11 that existing explicit association tasks conflate evaluation and interpretation, while our task circumvents this limitation. We hope that this clarifies what we see as an important benefit of our proposed methodology.

5. The author excludes all respondents who completed the elicitation task in less than half the median response time. Given the mixed recommendations in the methodological literature about excluding survey speeders from experimental samples, it would be helpful to know how sensitive the results are to this decision. Would retaining all respondents or those who completed the survey in more than, say, 25% of the median time alter the findings?

We thank the reviewer for pushing us to examine the robustness of our results. First, we want to clarify that, consistent with our pre-registration, “speeders” were only dropped in the correlational study—not in the experimental study. To test the robustness of the results from the correlational study to this exclusion criterion, we produced the following graphs, which demonstrate how the estimated regression coefficient for the relationship between imagined otherness and outgroup dehumanization changes when dropping participants on the basis of different possible response time thresholds. Along the x-axis, we plot different possible thresholds we might have used, including our pre-registered one. Zero on the x-axis corresponds to dropping no participants on the basis of their response time, while one-hundred on the x-axis corresponds to dropping all participants with a response time below the median response time. The y-axis plots, for each of these possible thresholds, the standardized regression coefficient (and 95% confidence intervals for that coefficient) when removing participants on the basis of the threshold indicated by the x-axis. Finally, the different graphs apply this procedure to different models from Table 2 (i.e., with different controls included in the models). As can be seen, a null association is never contained within the 95% confidence interval across all 300 regressions performed for this analysis.

6. The paper makes several comparisons between the types of schemas measured by the survey instrument, but the one comparison that is given the least attention in the main text is the relative congruence between personal and ingroup schemas. This is attended to in the appendix as a robustness check (those with stronger ingroup attachment exhibit greater congruence), but could this measure be used directly in the research design as well? My thinking was that personal-vs-ingroup schema congruence could be an interesting alternative measure of ingroup attachment to self-reported partisan identification (or ingroup identity in general). It could also serve as a theoretically interesting moderator of the link between imagined otherness and dehumanization. Leaving it to the supplemental robustness check section seems like a missed opportunity.

We thank the reviewer for these suggestions. We now perform these suggested analyses as “empirical extensions” of the findings in our correlational study. The full results are detailed in the appendix, but in brief our analyses confirm your intuition: We find that personal-ingroup divergence does significantly moderate the relationship between imagined otherness and outgroup dehumanization. We also point to following up on this analysis as a future direction for research on page 25. Thank you for suggesting this supplemental analysis, which adds nuance and richness to our main findings.

7. *The elicitation task introduced by the study seems like an effective measurement strategy for identifying the perceived schematic otherness of outgroup members. At the same time, the study also includes explicit self-reported measures of schematic otherness and uses them as a control in one of the regression models. Interestingly, this explicit measure is modestly correlated with the measure generated by the elicitation task, but it is even more highly correlated with dehumanization than the elicitation task measure. It would be helpful for the author to be clearer about the value added by the (clever but time-consuming) elicitation task above and beyond the self-reported measure. Why not just use the latter given its good performance in the models? What insights can be gleaned from the elicitation task that cannot be gleaned from the self-reported measure?*

We thank the reviewer (and other reviewers) for pushing us to consider the costs and benefits of our measurement strategy more carefully, especially relative to the more expedient, explicit measure we included as a control. We see three main advantages of our measurement technique. First, we believe that our main theoretical claim—that dehumanization can arise from considering the outgroup’s mind—is bolstered by the fact our procedure requires participants to complete a cognitively demanding, perspective-taking task. Second, our method offers a plausible approach to experimentally manipulate participants’ perceptions of similarities and differences (e.g., in the manner we do in Study 2) and thus examine the causal relationship between perceptions and outgroup attitudes. It would be much harder to replicate the design of Study 2 using a simple self-report measure without creating unintended demand effects. Finally, our scheme elicitation task allows us to examine the *content* associated with intergroup divisions in a way the self-report measure would not allow. To demonstrate this value, we’ve added an empirical extension (mentioned on pages 17-18 and detailed in the appendix) in which we examine the relative strength of imagined otherness based only on positive, neutral, and negative content. It was surprising to us, as we suspect it will be to other readers, that neutral content seems to have the most robust association with outgroup dehumanization. This is an insight that simply could not have emerged with the use of a simple self-report. Overall, we believe these results help demonstrate the unique value of our measurement approach, and we thank you for pushing us to explore this further.

8. *Dehumanization is measured using the Ascent of Man scale, as had been the case in prior psychological studies. I would have liked to see a fuller discussion of this instrument’s past validation, its advantages over alternative measures, and some of its limitations. I worry in particular that the images used in the scale are blunt (essentially asking respondents to compare specific groups to apes), evoke evolutionary theory (which some respondents may reject wholesale), and are inherently gendered (only males are depicted in the prompt). How might these characteristics of the instrument affect the measurement of dehumanization in the study? Did the author consider alternative measurement strategies?*

We agree with you that the Ascent of Man scale is problematic on several dimensions, even if it is one of the most widely used scales to assess dehumanization. In response to your comment, we add more discussion of the scale’s validation in previous studies, as well as our reasoning for using it in this particular study, on pages 13-14. On page 25, we also identify additional dehumanization measures that would be useful to explore in future research. On pages 13-14, we

also expand our discussion of the scale's limitations (including some of the points you raise above).

9. A few claims in the paper could use additional citations. In particular, the sociological research on dehumanization by Aliza Luft seems conspicuously missing from the references (and calls into question the statement that “sociological inquiry of dehumanization appears to have waned since [the post-war years]”). The theory section should engage in particular with Luft’s findings that dehumanization often follows from genocidal violence rather than preceding it. Another literature that’s frequently referenced in the paper but rarely actually cited is the work on affective polarization by scholars like Shanto Iyengar, Jamie Druckman, Yphtach Lelkes, Matthew Levendusky, and Lily Mason. Finally, for recent work on partisan differences in the conceptualization of America, see Sides, Tesler, and Vavreck’s 2018 book Identity Crisis and my 2021 AJS article, “The Partisan Sorting of ‘America’” (no pressure to cite the latter, of course, but it may offer some helpful references and findings to further justify the present test case).

We thank the reviewer for these helpful suggestions. We now include references to Aliza Luft’s important and relevant work in this vein (see page 3). We have also added additional citations to the affective polarization literature where appropriate (see pages 8-9 and 22-23). Finally, on page 9 we’ve added a mention of some of the work on different interpretations of “America” as a concept. We believe these additions have greatly improved the theoretical grounding for our contribution.

It was a pleasure to read this terrific piece of research. I look forward to recommending it to others and citing it myself in the future.

Thank you for your incredibly thoughtful and generous feedback!

RESPONSE TO REVIEWER 2

1) While there is an attempt to differentiate the main argument proposed in the article from existing research, the article lacks a proper literature review of the positions it sets itself apart from. There are some sections in which some of the relevant literature in sociology (though this seems to be a catch-phrase also including research on dehumanization in political science, political theory and philosophy) and (social) psychology is introduced though the article lacks a proper critical engagement with this literature to give the reader a sense of what the different positions are, how they are related to each other and how the current research sets itself apart from these existing positions. Such an outline would also benefit from a wider acknowledgement of dehumanization research and varying concepts of dehumanization (beyond institutional and cognitive accounts there are prominent positions which see dehumanization as a denial of moral status, as a discursive strategy or understand it behaviorally). While the article cannot offer a comprehensive review, it would benefit from a more balanced and substantial

consideration of the existing literature, not least as it would further emphasize the novelty of the approach offered here.

Thank you for pushing us to clarify our theoretical argument and contribution in the context of the broader literature on dehumanization. We have expanded our discussion of the literature on dehumanization (see pages 3-10), which we hope will help highlight how our theoretical approach is complementary to, but ultimately different from, what has come before it.

2) The article lacks a clear delineation of the concept of dehumanization. In the accounts it highlights at the beginning we find a clear sense of what dehumanization consists in (denial of mind, animalization, mechanization etc), i.e. these accounts establish clearly what they take to be the central component of 'humanity' that is being denied in cases of dehumanization. The article, however, does not offer such a conceptual delineation, so the reader is left wondering what dehumanization actually consists in and what element(s) of being human are actually denied in cases of 'imagined otherness'. At some point it seems that dehumanization describes the distance between a person's outgroup schema and the generalized other schema but then the article says that 'the tendency to dehumanize increases with the distance between these schemas' which implies that the discrepancy is distinct from dehumanization, so we still don't know what is meant by 'dehumanization'.

In response to your comment, we have revised our theoretical argument (see pages 3-10) to clarify what we mean by dehumanization. Although there are different conceptions of dehumanization in different literatures, we focus on a particular form: blatant dehumanization. We see imagined otherness not as a form of dehumanization but rather as a way of perceiving an outgroup as being fundamentally different from a "typical" human. We hope that our revisions have helped to clear up the confusion in the prior version of the manuscript and thank you for prompting us to make these changes.

Further along the article uses a slider scale linked to the 'ascent of man' model; even if this would be the way in which the article perceives dehumanization (ranking on a scale from ape to fully upright humans), the problem is that this scale was given to respondents as the only way to rate democrats and republicans – i.e. respondents did not choose to describe their perceptions of these groups on a ape to human scale. Whether their ranking actually expresses a belief that democrats/republicans are more ape-like (and hence less human) is unclear (we only know they rank them 'lower' but since the ascent of man ladder was the only way to express this, we cannot conclude that respondents substantially agreed with or would subscribe to expressing their ranking in relation to an ape-human categorization). Overall, the article needs to be much clearer as to the factor(s) central to the concept of 'human' that are being denied in order to justify using the concept of 'dehumanization' to describe the process it identifies.

We would first like to clear up a point of confusion about the evaluations that participants were asked to do. In addition to completing the Ascent of Man scale for Republicans and Democrats, participants were also asked to report their level of warmth toward both groups using feeling thermometers. In other words, they were able to rate the outgroup as "lower" in warmth even if they did not view them as more "ape-like" on the Ascent of Man scale. Thus, we believe that the Ascent of Man scale captured participants' tendency to dehumanize the outgroup independent of

their overall sentiment toward the outgroup. As to the central concept of “human,” we hope that our revised theory section (see pages 3-10) provides the clarity you were seeking.

3) Linked to the points above, it is also somewhat unclear what the connection between imagined otherness and dehumanization is. The article early on states that imagined otherness CAN arouse blatant dehumanization (page 2) with imagined otherness meaning “that a prototypical member of a social group construes an important facet of the social world in ways that diverge from the way most humans understand it”. It remains however, somewhat unclear, a) under what conditions imagined otherness DOES arouse blatant dehumanization (the stipulation that it CAN do so implies that it is not invariably the case) and b) what the ‘important facets’ are (it seems to imply that a group can construe (non-important) facets of the social world that do diverge from the way most humans understand them without leading to the process described here). The question then would be not only what these important facets are but also who determines them.

We agree that our use of “can” in our primary theoretical statements (including the title) was confusing and thank the reviewer for pushing us to change the wording (in the title and the text). With the addition of theoretical scope conditions (see pages 5 and 8), we also feel more comfortable using more declarative language.

We have also included a discussion of which concepts should count as “important facets” of the social world with respect to our theory on page 8. On pages 9-10, we describe why we chose to focus on the concept of “America” specifically. Finally, on page 25, we discuss how future research can examine the extent to which the mechanism of imagined otherness extends to concepts beyond “America.”

4) Finally, while the article provides some reflections on the consequences of its research specifically in terms of political polarization in the US, it remains somewhat vague as to the extent blatant dehumanization is linked with (political) behavioural outcomes, an aspect that much of dehumanization research is focused on (e.g. is the presence of dehumanization in US political discourse linked to increasingly extreme forms of political activism?). I am not sure if the article can or does want to make any claims in this respect but it would be interesting to know whether this research can further contribute to the link between dehumanizing cognitive dispositions and behavioural outcomes (especially since this link has come under increasing criticism in recent studies – e.g. Harriet Over, Johannes Lang, Nicolas Mariot).

We thank the reviewer for inviting us to speculate on the potential consequences of cross-partisan dehumanization. We have added a discussion to this effect on pages 22-23.

Overall, I think this article presents an interesting and novel contribution to existing research on dehumanization though it would benefit from a more critical and substantive engagement with the existing literature, a clearer conceptual delineation of what is meant by dehumanization and a more explicit discussion of the link between imagined otherness and the occurrence of dehumanization.

Thank you for your insightful and constructive feedback!

RESPONSE TO REVIEWER 3

First, the introduction seems a little scattered. Why would “imagined otherness” elicit blatant dehumanization? Is it because that imagined otherness is linked with a “diminished mind”, or because it implies a distinct, and “less than human” essence? The authors talked about both accounts but failed to provide a clear explanation.

We have re-written the theoretical formulation (see pages 3 - 10) in the hopes of clarifying these points. We hope that the process we hypothesize is now clearer.

A related question is, are all the social perceptions people have a core element of mind or human essence? The authors used the specific concept of “America” as the focal word, which I think, may not (but should be) explicitly stated. Can the results be generalized to other concepts such as “democracy”, “country”, “freedom”, “love”...? Or, do the concepts we should use in the test depend on the particular outgroup we are evaluating? This is a vital question the authors should think about because it is not only closely tied to the robustness and generalizability of the results, but also of crucial theoretical importance.

In response to your comment, we have included a discussion of which concepts should count as “important facets” of the social world with respect to our theory on page 8. On pages 9-10, we describe why we chose to focus on the concept of “America” specifically. Finally, on page 25, we discuss how future research can examine the extent to which the mechanism of imagined otherness extends to concepts beyond “America.”

The authors defined “imagined otherness” as the outgroup-generalized other schematic difference, which does make sense. But how about the personal-outgroup and ingroup-outgroup differences? Are they similarly associated with dehumanization? After all, people often see themselves and ingroup as more human than others. I think it doesn't hurt to also report these results.

We agree that both personal-outgroup divergence (incongruity between one's own schema and one's attribution of the outgroup's schema) and intergroup divergence (incongruity between one's attribution of the ingroup's schema and one's attribution of the outgroup's schema) are plausible alternatives to imagined otherness in our theory. In Table 2, we report the results of OLS models predicting dehumanization from these measures and imagined otherness simultaneously. Results show that while imagined otherness is strongly significant in these models, personal-outgroup divergence and intergroup divergence are not.

More demographic information about the participants may be given.

We have added the only other piece of demographic information (religion) that we have about the sample in our correlational study. Unfortunately, we are unable to access detailed demographic information for our experimental study.

The authors reported a more straightforward measure of perceived otherness, which they named self-reported othering. By asking people how much they believe an outgroup's social perceptions are different from the perceptions of generalized others, we can know directly how aberrant the outgroup's thinking is thought to be. I noticed that this index showed a greater association with blatant dehumanization than the measure of imagined otherness. Then why bother devising the schema elicitation task? Although the task seems more objective than the subjective report, it does have many limitations. First, we need to choose an appropriate focal word (Or does any word work? I'm afraid not..). Second, it seems very laborious and bothersome to generate the many related concepts. Third, we have to force the respondents to choose from the predetermined concepts. Subjectivity may be introduced in all the steps. Instead, with the subjective report, we may have a more comprehensive understanding of perceived otherness by asking participants how large are the differences in perceiving concepts such as "human", "America", "country", "freedom", "morality", "love"...by a certain outgroup and by a typical person! In a word, I do have serious concerns about the measure.

We found this comment to be very in-line with point 7 of Reviewer 1's comments. To avoid repetition, may we refer you to our response to this point above?

I know it's beyond the scope of this research but do wonder if the reversed causal link also holds. If we dehumanize an outgroup more, would we see a larger schematic distance between them and ourselves? It seems to make sense too.

We thank the reviewer for inviting us to speculate on the existence of a causal relationship in the reverse direction. While our empirical results don't allow us to speak to the existence of such a relationship, we believe it's important to include a discussion of it in the manuscript on page 25. We have added this and hope it stimulates future research.

Although the studies focus on blatant dehumanization, the authors may also measure and report subtle dehumanization to examine whether imagined otherness uniquely predicts blatant dehumanization.

In response to your comment, we now highlight on page 25 of the discussion some additional measures of dehumanization that would be fruitful to explore in future research.

In sum, I really like the idea the authors proposed. Yet, I don't think they have made it clear why imagined otherness leads to blatant dehumanization, or whether the measure of imagined otherness is valid or necessary.

Thank you for your incredibly helpful comments! We hope that our revisions have resolved any confusion and assuaged any concerns.

27th Nov 23

Dear Dr van Loon,

Your manuscript titled "Imagined Otherness: Perceived Schematic Difference Fuels Dehumanization" has now been seen by our reviewers, whose comments appear below. In light of their advice I am delighted to say that we are happy, in principle, to publish a suitably revised version in Communications Psychology under the open access CC BY license (Creative Commons Attribution v4.0 International License).

We therefore invite you to revise your paper one last time to address the remaining concerns of our reviewers and a list of editorial requests. At the same time we ask that you edit your manuscript to comply with our format requirements and to maximise the accessibility and therefore the impact of your work.

Please note that it may still be possible for your paper to be published before the end of 2023, but in order to do this we will need you to address these points as quickly as possible so that we can move forward with your paper.

EDITORIAL REQUESTS:

SUBMISSION INFORMATION:

OPEN ACCESS:

Communications Psychology is a fully open access journal. Articles are made freely accessible on publication under a CC BY license (Creative Commons Attribution 4.0 International License). This license allows maximum dissemination and re-use of open access materials and is preferred by many research funding bodies.

For further information about article processing charges, open access funding, and advice and support from Nature Research, please visit <https://www.nature.com/commspsychol/article-processing-charges>

At acceptance, you will be provided with instructions for completing this CC BY license on behalf of all authors. This grants us the necessary permissions to publish your paper. Additionally, you will be

asked to declare that all required third party permissions have been obtained, and to provide billing information in order to pay the article-processing charge (APC).

* **DATA AVAILABILITY:**

[link redacted]

Best regards,

Jennifer Bellingtier

Jennifer Bellingtier, PhD
Senior Editor
Communications Psychology

REVIEWERS' EXPERTISE:

Reviewer #1 dehumanization, sociology

Reviewer #2 dehumanization, group dynamics

Reviewer #3 dehumanization, social cognition

REVIEWERS' COMMENTS:

Reviewer #1 (Remarks to the Author):

Having closely read the revision memo and the new version of the manuscript, I am fully satisfied with the authors' response to my review. I appreciate their thoughtful engagement with my critiques, the additional clarifications provided in the paper, and the supplemental analyses offered in the memo and appendices. In my view, the revised paper is considerably richer and more compelling than the initial submission, which is what one would hope for after a successful peer review process. I have no further concerns at this point.

-BB

Reviewer #2 (Remarks to the Author):

The revised article shows that the authors have constructively engaged with my comments. My main points of concerns have been addressed sufficiently, so I would recommend acceptance.

Torsten Michel

Reviewer #3 (Remarks to the Author):

I'm convinced by the idea behind the research and believe that imagined otherness may be linked with dehumanization of outgroups. This said, I am still conservative about the operationalization of the construct as well as its practical value. The operationalization may be more like a theoretical illustration, rather than a measure one can really put to application. There is still much to be clarified and refined.